# CBP: Learning Shared Cognitive Basis Space and Connectivity Patterns for Cross-Task Brain Dynamics Modeling

## Abstract

Understanding the brain dynamics underlying human cognition requires models that jointly achieve modeling capability and interpretability across tasks. Existing approaches either rely on deep learning models, which learn strong but implicitly encoded representations that are difficult to interpret, or explicit state-space models, which are interpretable but limited to single-task settings. To bridge this gap, we present CBP, an interpretable, non–deep-learning framework for cross-cognitive-task brain modeling. By explicitly leveraging shared information across tasks, CBP accurately recovers latent components while maintaining state-of-the-art performance. Importantly, CBP uncovers a stable set of **C**ognitive **B**ases and connectivity **P**atterns in the human brain. The reliability of these discoveries is supported by extensive quantitative evaluations and a battery of perturbation analyses, demonstrating their robustness. Moreover, leveraging these shared cognitive components allows CBP to generalize effectively to both the HCP and multi-stage learning datasets, where it not only achieves accurate prediction of task states and learning outcomes but also reveals the key cognitive connectivity patterns underlying these behaviors. Together, these results establish CBP as an interpretable framework for large-scale cognitive dynamics, offering mechanistic insight into cross-task cognition. The code is available at https://anonymous.4open.science/r/CBP-2121.

## 1 Introduction

Understanding the neural dynamics underlying complex cognitive functions remains a core challenge in neuroscience. Addressing this problem is crucial for uncovering the principles of human cognition, explaining brain network organization, and advancing psychiatric diagnosis and brain–computer interfaces Buonomano & Maass (2009); Miller & Wilson (2008). Functional magnetic resonance imaging (fMRI) enables non-invasive, whole-brain recordings of blood-oxygen-level-dependent signals, allowing researchers to model brain dynamics during cognitive tasks. Yet most existing approaches focus on single-task settings and lack a unified framework to capture task-general neural dynamics. This gap presents two key challenges: i) The pronounced heterogeneity of neural dynamics across tasks makes unified modeling difficult, demanding models that are both robust and capable of generalizing under limited-data and novel-task conditions. ii) Interpretability is critical so that model parameters can be directly linked to task variables and empirically validated. However, current approaches fall short of these goals.

Current approaches to brain dynamics modeling fall into two main categories: (i) Deep learning models. Brain dynamics models, such as NetFormer Lu et al. (2025), LtrRNN Pellegrino et al. (2023), AMAG Li et al. (2023), HGFM Han et al. (2025), and Brain-MoE Wei et al. (2025), can capture rich spatiotemporal structure and learn powerful latent representations. However, they typically encode neural dynamics implicitly within high-dimensional parameters, producing distributed and difficult-to-interpret representations. **In neuroscience-oriented applications, interpretability is essential rather than optional Mudrik et al. (2025), yet the internal computations of deep neural networks offer limited transparency and provide little mechanistic insight into the underlying neural processes.** (ii) Explicit state-space models. Approaches such as SLDS Linderman et al. (2016); Glaser et al. (2020) and

dLDS Mudrik et al. (2025; 2024a) provide interpretable latent states and transition matrices but remain restricted to single-task settings, failing to capture shared dynamics across tasks.

Overall, existing methods cannot simultaneously achieve predictive performance and mechanistic interpretability in multi-cognitive-task settings, highlighting the need for new modeling frameworks (Fig. 1).

To address this gap, **we hypothesize that human brain dynamics across cognitive tasks are organized by a set of global, temporally invariant shared components.** This is inspired by recent mouse studies, which reveal that large-scale cortical dynamics collapse onto a finite set of stable, low-dimensional spatiotemporal patterns that persist across behavioral states Mac-Dowell & Buschman (2020); MacDowell et al. (2024; 2025). To test the hypothesis, we introduce a method that learns a shared basis space and connectivity patterns to model and interpret cross-task brain dynamics. Within this framework, the heterogeneity of task-evoked dynamics is no longer treated as an obstacle but reinterpreted as an opportunity: neural activity from different tasks is viewed as distinct trajectories of a unified cognitive system, representing task-specific unfoldings of the shared basis and its interactions. This enables us to exploit shared information across tasks, boosting both modeling accuracy and cross-task generalization. Our main contributions are as follows:

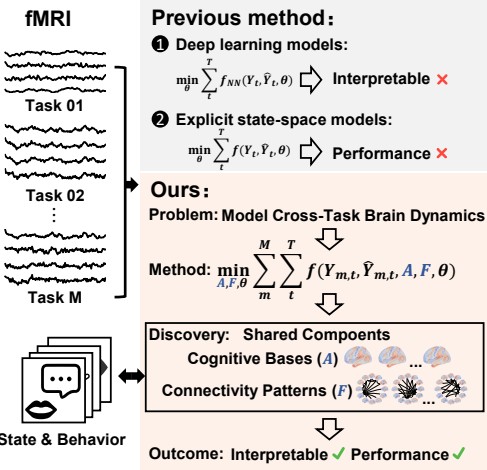

Figure 1: **Motivation.** (Top): The limitations of previous methods. (Bottom): Our model overcomes these limitations by learning shared cognitive bases and their connectivity patterns.

- **Methodology:** We present CBP, an interpretable, non–deep-learning framework for cross-task brain modeling. It exploits shared information across tasks and accurately recovers latent components while maintaining state-of-the-art performance.

- **Scientific Discovery:** CBP uncovers a stable set of cognitive bases and connectivity patterns in humans, supporting the core hypothesis. These findings are validated through multiple quantitative metrics and diverse perturbation tests.

- **Generalization:** CBP, grounded in shared cognitive components, generalizes well to the HCP and multi-stage learning datasets. It accurately predicts task states and learning outcomes and reveals the key cognitive connectivity patterns that drive these behaviors.

## 2 RELATED WORK

**Cognitive Component Hypothesis:** Recent mouse studies reveal that large-scale cortical activity is organized by a finite set of low-dimensional, stable spatiotemporal motifs, proposed as cognitive components. MacDowell and Buschman (MacDowell & Buschman (2020)) identified 14 reproducible motifs across mice and task conditions, explaining most cortical variance. MacDowell et al. (MacDowell et al. (2024)) further demonstrated that typical and atypical behavioral phenotypes share 16 common motifs but differ in expression frequency, which predicts functional connectivity and behavior. More recently, MacDowell et al. (MacDowell et al. (2025)) proposed a shared-subspace model, showing that cortical regions dynamically align across subspaces to route motifs into different networks, enabling cognitive flexibility. Together, these findings suggest that cortical computation relies on a finite set of reusable components, with cognitive differences arising from their dynamic selection and composition. However, this hypothesis remains untested in large-scale human cognitive dynamics. Here, we reveal a set of global, temporally invariant cognitive basis spaces and connectivity patterns in the human brain, providing direct evidence for human cognitive shared components.

**Existing Approaches to Model Cross-Task Brain Dynamics:** Approaches to modeling brain dynamics broadly fall into two categories: **i) Deep learning models:** Deep learning models for brain dynamics span a broad range of architectures, including recurrent models such as LtrRNN Pellegrino et al. (2023), Transformer-based frameworks such as NetFormer Lu et al. (2025), graph-based approaches such as AMAG Li et al. (2023), and large-scale representation models such as HGFM Han et al. (2025) and Brain-MoE Wei et al. (2025). These methods capture complex spatiotemporal structure and learn high-capacity latent representations that enhance predictive performance. **However, they encode neural dynamics implicitly through end-to-end optimization over large parameter spaces, producing highly distributed and difficult-to-interpret representations. This black-box nature limits the ability to relate their internal computations or predictions to specific neural components, which is essential for neuroscience-oriented interpretability.** **ii) Explicit state-space models:** Another line of work uses switching linear dynamical systems (SLDS) (Linderman et al. (2016; 2017); Glaser et al. (2020)), which capture distinct dynamical regimes via discrete state transitions but struggle to represent multiple simultaneously active processes. Decomposed LDS (dLDS) and its extensions (Mudrik et al. (2025; 2024b); Chen et al. (2024); Mudrik et al. (2024a)) generalize the switching assumption into a sparse, time-varying decomposition to identify co-activated subcircuits. **Nonetheless, SLDS and dLDS remain largely confined to single-task settings and fail to capture task-general dynamical components.** To overcome these limitations, we propose an interpretable, non–deep-learning framework that uncovers scientifically meaningful cognitive bases and connectivity patterns shared across diverse cognitive tasks.

## 3 PROBLEM DESCRIPTION

Let $\{Y_m^{(s)}\}_{m=1}^M$ denote the collection of fMRI recordings acquired from subject $s$ across $M$ cognitive tasks (the *Over 100 Task* (OT) dataset). For task $m$, the neural activity is represented as $Y_m^{(s)} \in \mathbb{R}^{N \times T_m^s}$, where $N$ denotes the number of brain regions, $T_m^s$ is the number of time points. Each row of $Y_m^{(s)}$ represents the time series of a specific brain region. Due to the inherent diversity in functional demands across tasks, the dynamical patterns $Y_m^{(s)}$ and $Y_{m'}^{(s)}$ exhibit substantial heterogeneity, with each task revealing a distinct dynamic manifestation of the brain's cognitive system under specific conditions. Moreover, inter-subject variability and trial-to-trial fluctuations further exacerbate this heterogeneity. A central challenge lies in developing a unified model capable of accurately reconstructing brain dynamics across diverse cognitive tasks, while providing insights into the mechanisms that govern cognitive processes.

## 4 HYPOTHESIS AND MODELING GOALS

We assume that the cognitive brain system is composed of $K$ fundamental components (referred to as **cognitive bases**: $A_{:i} \in \mathbb{R}^N, i = 1, ..., K$). The observed signal is modeled as the collective activity of these bases: $Y_m^{(s)} = A X_m^{(s)} + \eta$, where $A \in \mathbb{R}^{N \times K}$ is the basis space matrix, $X_m^{(s)} \in \mathbb{R}^{K \times T_m^s}$ the latent trajectories, and $\eta$ i.i.d. Gaussian noise. Since the brain is an inherently dynamical system, these cognitive bases are dynamically activated and interact with one another during task execution. We assume that such interactions, within each task, follow nonlinear and nonstationary dynamics: $x_{m,t}^{(s)} = F_{m,t}^{(s)}(x_{m,t-1}^{(s)})$, $F_{m,t}^{(s)} : \mathbb{R}^K \to \mathbb{R}^K$. $x_{m,t}^{(s)} \in \mathbb{R}^K$ is the $t$-th column of the matrix $X_m^{(s)} \in \mathbb{R}^{K \times T_m^s}$. However, directly solving such $F_{m,t}^{(s)}$ is intractable, as at each time step one would need to resolve $\arg\max_{F_{m,t}^{(s)}} p(F_{m,t}^{(s)} \mid x_{m,t}^{(s)}, x_{m,t-1}^{(s)})$, with far more unknowns than equations. Inspired by the dLDS framework Mudrik et al. (2024a), we assume these dynamics with a set of **connectivity patterns** that are globally shared and invariant across tasks: $\{f_p\}_{p=1}^P$, $f_p \in \mathbb{R}^{K \times K}$. Each pattern encodes a core interaction among cognitive bases and is reused across different segments of the system's trajectory. These patterns can capture both global, time-invariant interactions and task-specific variations through their time-dependent activations. Multiple patterns may be active concurrently or intermittently, enabling a unified mechanism to model the diverse neural trajectories observed across tasks flexibly.

Therefore, **CBP** leverages multi-cognitive-task observational data $\{Y_m\}_{m=1}^M$ to model cross-task brain dynamics and achieve two goals: i) Identify a set of shared cognitive bases and their connec-

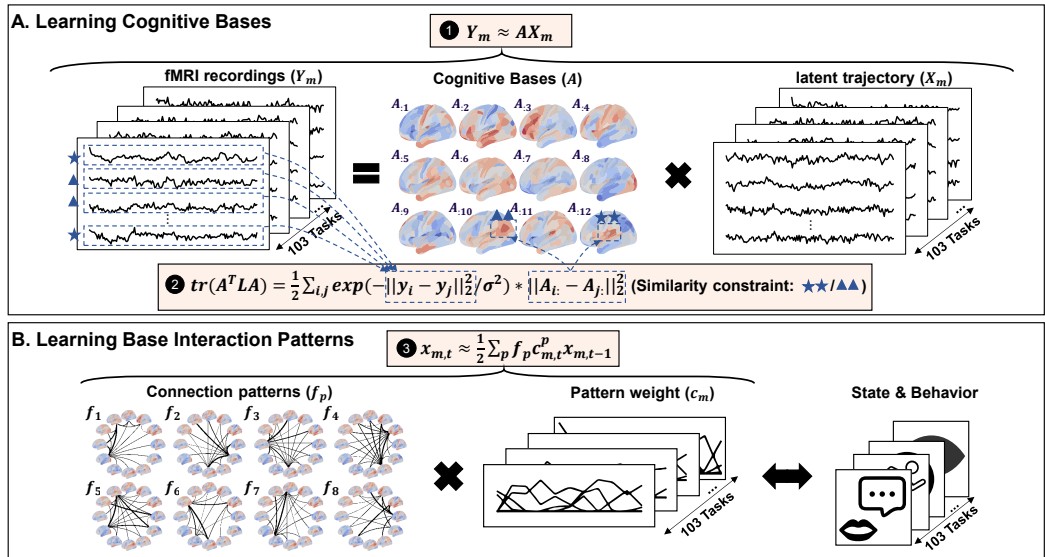

Figure 2: **Framework of CBP. (A) Learning Cognitive Bases:** fMRI recordings $Y_m$ are factorized into a set of shared cognitive bases $A$ and latent trajectories $X_m$. The manifold regularization term $\mathrm{tr}(A^\top LA)$ extracts important information from the data, enabling a task-invariant representation of the shared cognitive basis space. **(B) Learning Basis connectivity patterns:** The latent trajectories $X_m$ are further modeled through a set of global connectivity patterns $\{f_p\}_{p=1}^P$ and their activation coefficients $c_{m,t}^p$. This decomposition disentangles stable patterns from a 103-task fMRI dataset and provides a unified modeling for cross-task brain dynamics.

tivity patterns; ii) Elucidate how these patterns jointly and independently drive interactions among cognitive bases, thereby explaining the dynamical mechanisms underlying cognition in the brain.

## 5 METHOD

Our approach is designed to simultaneously model the global cognitive basis space across multiple tasks and the connectivity patterns that underlie interactions among cognitive bases. It is predicated on two key assumptions: i) a shared basis space can be identified across different tasks, and ii) the full repertoire of interactions among bases can be spanned by a global set of linear dynamical systems, whose linear combinations evolve over time to capture non-stationary dynamics. Building on these assumptions, we propose CBP, whose objective function is formulated as:

$$
\min_{\{A, X_m^{(s)}, f_p, c_{m,t}^{p,(s)}\}} \sum_{s=1}^{S} \sum_{m=1}^{M} \left[ \|Y_m^{(s)} - AX_m^{(s)}\|_2^2 + \sum_{t=2}^{T_m^s} \lambda_x \left\| x_{m,t}^{(s)} - \sum_{p=1}^{P} f_p c_{m,t}^{p,(s)} x_{m,t-1}^{(s)} \right\|_2^2 \right.
$$
$$
\left. + \lambda_a \mathrm{tr}(A^\top LA) + \beta(f_p, c_{m,t}^{p,(s)}, A) \right]. \tag{1}
$$

The first and second terms aim to encourage the precise reconstruction of observed data and latent dynamics, where the basis space matrix $A$ and the set of dynamic functions $\{f_p\}_{p=1}^P$ capture shared information across different tasks. In this way, they effectively learn common features across tasks, thereby enhancing the modeling of cross-task dynamics.

The third term is the manifold regularization term: $tr(A^\top LA)$, where $L$ is the Laplacian matrix defined as $L = D - W$, with $W$ being the similarity matrix that measures the pairwise similarity between brain regions. Specifically, the similarity between brain regions $i$ and $j$ is defined as: $W_{ij} = \exp\left(-\frac{\|y_i - y_j\|_2^2}{\sigma^2}\right)$, $i \neq j$, where $y_i$ and $y_j$ represent the time series of the $i$-th and $j$-th brain re-

gions, respectively. The parameter $\sigma$ controls the scale of the similarity measure. Self-connections are set to $W_{ii} = 0$. The regularization term $tr(A^\top LA) = \frac{1}{2}\sum_{i,j}\exp\left(-\frac{\|y_i-y_j\|_2^2}{\sigma^2}\right)\cdot\|A_{i:}-A_{j:}\|_2^2$ has the following two advantages: i) High-dimensional data processing: fMRI data is typically high-dimensional, with each scan involving signals from hundreds of brain regions, increasing the complexity and noise of the data. Manifold regularization helps by constraining local structures, reducing unnecessary high-dimensional noise, and extracting important signals from the data, while ignoring redundant noise or irrelevant features. ii) Cross-task and cross-subject generalization: fMRI signals are often collected across multiple tasks and subjects, and the signals from each task or subject may exhibit heterogeneity. Manifold regularization allows for learning a shared low-dimensional structure $A$ across different tasks and subjects, rather than relying on the independent features of each task or subject, thus improving generalization across tasks and subjects.

The fourth term $\beta(f_p, c_{m,t}^{p,(s)}, A) = \sum_{t=2}^{T_m^s}\lambda_c\|c_{m,t}^{p,(s)} - c_{m,t-1}^{p,(s)}\|_2^2 + \lambda_\rho\sum_{p_1,p_2,\,(p_1\neq p_2)}\rho(f_{p_1}, f_{p_2}) + \sum_{t=1}^{T_m^s}\left(\alpha_1\|c_{m,t}^{p,(s)}\|_1 + \alpha_2\|f_p\|_1 + \alpha_3\|A\|_1\right)$ controls the temporal smoothness of the dynamic coefficients $c_{m,t}^{p,(s)}$, decorrelation across different connectivity patterns $f_p$, and sparsity of both dynamic coefficients $c_{m,t}^{p,(s)}$ and shared components, where $c_{m,t}^{p,(s)}$ denotes the activation coefficient of the $p$-th connectivity pattern $f_p$ at time $t$ for the $m$-th cognitive task of subject $s$, $\rho(f_{p_1}, f_{p_2}) = \left(\frac{\langle\text{vec}(f_{p_1}),\,\text{vec}(f_{p_2})\rangle}{\|\text{vec}(f_{p_1})\|_2\,\|\text{vec}(f_{p_2})\|_2}\right)^2$, $\text{vec}(.)$ denotes the vectorization operator that flattens a matrix into a vector. Therefore, the term $\beta(.)$ enhances the stability, interpretability, and generalization ability of the model. To optimize this objective, we adopt the alternating convex search (ACS) algorithm to estimate $A$, $X_m^{(s)}$, $c_{m,t}^{p,(s)}$ and $\{f_p\}_{p=1}^P$.

**Cognitive Bases Update:** Based on the data from 103 cognitive tasks, we identify the shared basis matrix $A$, which can be updated as follows:

$$\hat{A} = \arg\min_A \sum_{s=1}^{S}\sum_{m=1}^{M}\|Y_m^{(s)} - AX_m^{(s)}\|_2^2 + \lambda_a tr(A^\top LA) + \alpha_3\|A\|_1. \tag{2}$$

Here, $\lambda_a$ controls the manifold regularization, and $\alpha_3$ controls the sparsity regularization on the basis space. Together, these terms encourage $A$ to capture task-invariant shared information, thereby improving cross-task generalization.

**Latent State and Dynamic Coefficients Update:** The latent trajectories $X_m^{(s)}$ and the activation weights $c_{m,t}^{p,(s)}$ of the connectivity patterns are iteratively updated as follows.

$$\hat{X}_m^{(s)}, \hat{c}_{m,t}^{p,(s)} = \arg\min_{X_m^{(s)},c_{m,t}^{p,(s)}} \sum_{s=1}^{S}\sum_{m=1}^{M}\left[\|Y_m^{(s)} - AX_m^{(s)}\|_2^2 + \sum_{t=2}^{T_m^s}\left(\lambda_x\left\|x_{m,t}^{(s)} - \sum_{p=1}^{P}f_p c_{m,t}^{p,(s)} x_{m,t-1}^{(s)}\right\|_2^2\right.\right.$$
$$\left.\left. + \lambda_c\|c_{m,t}^{p,(s)} - c_{m,t-1}^{p,(s)}\|_2^2\right) + \sum_{t=1}^{T_m^s}\alpha_1\|c_{m,t}^{p,(s)}\|_1\right], \tag{3}$$

where $\lambda_c$ regulates temporal smoothness, constraining the trajectories to evolve in a stable and continuous manner, while $\alpha_1$ controls the sparsity of connectivity pattern activations, encouraging only a small subset to be active at each time point.

**Connectivity Patterns Update:** The connectivity patterns, which characterize the interactions among cognitive bases, are identified as follows:

$$\hat{f}_p = \arg\min_{f_p} \sum_{s=1}^{S}\sum_{m=1}^{M}\left[\sum_{t=2}^{T_m^s}\lambda_x\left\|x_{m,t}^{(s)} - \sum_{p=1}^{P}f_p c_{m,t}^{p,(s)} x_{m,t-1}^{(s)}\right\|_2^2\right.$$
$$\left. + \alpha_2\|f_p\|_1 + \lambda_\rho\sum_{p_1,p_2,\,(p_1\neq p_2)}\rho(f_{p_1}, f_{p_2})\right], \tag{4}$$

where $\alpha_2$ promotes sparsity within the connectivity patterns, while $\lambda_\rho$ prevents different $f_p$ from becoming overly similar. The framework of CBP is shown in Fig. 2 and its algorithm is presented

Table 1: Performance comparison (MSE and Pearson correlation) of CBP and baseline models on multiple datasets for reconstructing the observed signals $Y$ and identifying the cognitive bases $A$, latent trajectories $X$.

| Dataset | Three-Task Synthetic Dataset | | | Ten-Task Synthetic Dataset | | | OT Dataset | HCP Dataset | TLAE Dataset |
|---|---|---|---|---|---|---|---|---|---|
| Metric | $MSE(Y,\hat{Y})$ | $MSE(A,\hat{A})$ | $MSE(X,\hat{X})$ | $MSE(Y,\hat{Y})$ | $MSE(A,\hat{A})$ | $MSE(X,\hat{X})$ | $MSE(Y,\hat{Y})$ | $MSE(Y,\hat{Y})$ | $MSE(Y,\hat{Y})$ |
| SLDS | 0.0299 ± 0.0002 | 0.1645 ± 0.0005 | 1.4646 ± 0.0058 | 0.0778 ± 0.0006 | 0.0618 ± 0.0004 | 1.2111 ± 0.0081 | 0.0126 ± 0.0010 | 0.0095 ± 0.0005 | 0.0051 ± 0.0004 |
| rSLDS | 0.0257 ± 0.0003 | 0.1619 ± 0.0007 | 1.2293 ± 0.0024 | 0.0774 ± 0.0004 | 0.0570 ± 0.0007 | 1.4246 ± 0.0064 | 0.0034 ± 0.0004 | 0.0085 ± 0.0005 | 0.0594 ± 0.0021 |
| mp-rSLDS | 0.0199 ± 0.0004 | 0.0409 ± 0.0005 | 0.0891 ± 0.0005 | 0.0574 ± 0.0001 | 0.0621 ± 0.0001 | 1.2286 ± 0.0065 | 0.0262 ± 0.0016 | 0.0077 ± 0.0008 | 0.0046 ± 0.0006 |
| CBP | **0.0052** ± 0.0002 | **0.0041** ± 0.0003 | **0.0095** ± 0.0002 | **0.0021** ± 0.0003 | **0.0081** ± 0.0002 | **0.0092** ± 0.0002 | **0.0016** ± 0.0003 | **0.0075** ± 0.0002 | **0.0023** ± 0.0002 |

| Dataset | Three-Task Synthetic Dataset | | | Ten-Task Synthetic Dataset | | | OT Dataset | HCP Dataset | TLAE Dataset |
|---|---|---|---|---|---|---|---|---|---|
| Metric | $\rho(Y,\hat{Y})$ | $\rho(A,\hat{A})$ | $\rho(X,\hat{X})$ | $\rho(Y,\hat{Y})$ | $\rho(A,\hat{A})$ | $\rho(X,\hat{X})$ | $\rho(Y,\hat{Y})$ | $\rho(Y,\hat{Y})$ | $\rho(Y,\hat{Y})$ |
| SLDS | 0.7684 ± 0.0012 | 0.8109 ± 0.0002 | 0.7684 ± 0.0006 | 0.7909 ± 0.0007 | 0.2351 ± 0.0015 | 0.4698 ± 0.0008 | 0.6322 ± 0.0002 | 0.6188 ± 0.0006 | 0.7157 ± 0.0008 |
| rSLDS | 0.7911 ± 0.0002 | 0.8141 ± 0.0004 | 0.5871 ± 0.0011 | 0.8064 ± 0.0004 | 0.7870 ± 0.0010 | 0.6460 ± 0.0005 | 0.8657 ± 0.0005 | 0.7175 ± 0.0004 | 0.6933 ± 0.0003 |
| mp-rSLDS | 0.8428 ± 0.0007 | 0.8403 ± 0.0006 | 0.6080 ± 0.0006 | 0.7951 ± 0.0003 | 0.5556 ± 0.0007 | 0.4512 ± 0.0003 | 0.7297 ± 0.0002 | 0.7024 ± 0.0005 | 0.6082 ± 0.0003 |
| CBP | **0.9711** ± 0.0002 | **0.9403** ± 0.0003 | **0.9925** ± 0.0002 | **0.9478** ± 0.0003 | **0.8875** ± 0.0004 | **0.9678** ± 0.0003 | **0.9116** ± 0.0002 | **0.7617** ± 0.0003 | **0.9067** ± 0.0002 |

Table 2: Ablation Analysis

| Dataset | Three-Task Synthetic Dataset | | | | Ten-Task Synthetic Dataset | | | | OT Dataset |
|---|---|---|---|---|---|---|---|---|---|
| Metric | $MSE(Y,\hat{Y})$ | $MSE(A,\hat{A})$ | $MSE(X,\hat{X})$ | $MSE(F,\hat{F})$ | $MSE(Y,\hat{Y})$ | $MSE(A,\hat{A})$ | $MSE(X,\hat{X})$ | $MSE(F,\hat{F})$ | $MSE(Y,\hat{Y})$ |
| CBP.a | 0.0214 ± 0.0002 | 0.0417 ± 0.0004 | 1.8403 ± 0.0015 | 0.3589 ± 0.0012 | 0.0066 ± 0.0004 | 0.0468 ± 0.0006 | 0.0253 ± 0.0004 | 0.1961 ± 0.0004 | 0.0092 ± 0.0006 |
| CBP.b | 0.0235 ± 0.0004 | 0.0831 ± 0.0002 | 3.5067 ± 0.0012 | 0.8292 ± 0.0016 | 0.0168 ± 0.0004 | 0.0601 ± 0.0002 | 0.0743 ± 0.0004 | 0.1216 ± 0.0011 | 0.0095 ± 0.0004 |
| CBP.c | 0.0696 ± 0.0008 | 0.0406 ± 0.0004 | 0.5625 ± 0.0016 | 0.6710 ± 0.0014 | 0.0144 ± 0.0006 | 0.0456 ± 0.0004 | 0.3551 ± 0.0004 | 0.0692 ± 0.0002 | 0.0032 ± 0.0002 |
| CBP.d | 0.0260 ± 0.0002 | 0.0462 ± 0.0004 | 2.0331 ± 0.0018 | / | 0.0119 ± 0.0004 | 0.0458 ± 0.0008 | 0.7999 ± 0.0011 | / | 0.0090 ± 0.0003 |
| CBP.e | 0.0122 ± 0.0004 | 0.1087 ± 0.0006 | 1.4564 ± 0.0012 | / | 0.0062 ± 0.0002 | 2.3308 ± 0.0016 | 0.1370 ± 0.0004 | / | 0.3123 ± 0.0008 |
| CBP | **0.0052** ± 0.0002 | **0.0041** ± 0.0003 | **0.0095** ± 0.0002 | **0.0075** ± 0.0002 | **0.0021** ± 0.0003 | **0.0081** ± 0.0002 | **0.0092** ± 0.0002 | **0.0127** ± 0.0003 | **0.0016** ± 0.0003 |

| Dataset | Three-Task Synthetic Dataset | | | | Ten-Task Synthetic Dataset | | | | OT Dataset |
|---|---|---|---|---|---|---|---|---|---|
| Metric | $\rho(Y,\hat{Y})$ | $\rho(A,\hat{A})$ | $\rho(X,\hat{X})$ | $\rho(F,\hat{F})$ | $\rho(Y,\hat{Y})$ | $\rho(A,\hat{A})$ | $\rho(X,\hat{X})$ | $\rho(F,\hat{F})$ | $\rho(Y,\hat{Y})$ |
| CBP.a | 0.8646 ± 0.0003 | 0.4568 ± 0.0006 | 0.7270 ± 0.0003 | 0.8089 ± 0.0004 | 0.1994 ± 0.0004 | | 0.8250 ± 0.0003 | 0.8511 ± 0.0004 | 0.6161 ± 0.0004 |
| CBP.b | 0.8615 ± 0.0004 | 0.0284 ± 0.0012 | 0.3300 ± 0.0008 | 0.6167 ± 0.0002 | 0.4324 ± 0.0006 | 0.0951 ± 0.0008 | 0.8548 ± 0.0004 | 0.9274 ± 0.0003 | 0.6105 ± 0.0002 |
| CBP.c | 0.4886 ± 0.0005 | 0.4478 ± 0.0006 | 0.3589 ± 0.0010 | 0.4951 ± 0.0009 | 0.5514 ± 0.0008 | 0.2232 ± 0.0011 | 0.0341 ± 0.0006 | 0.9623 ± 0.0004 | 0.8152 ± 0.0003 |
| CBP.d | 0.8453 ± 0.0002 | 0.0056 ± 0.0012 | 0.1379 ± 0.0006 | / | 0.6501 ± 0.0004 | 0.0192 ± 0.0012 | 0.1748 ± 0.0005 | / | 0.6148 ± 0.0004 |
| CBP.e | 0.8885 ± 0.0004 | 0.5435 ± 0.0006 | 0.1538 ± 0.0011 | / | 0.8377 ± 0.0004 | 0.2347 ± 0.0007 | 0.2964 ± 0.0008 | / | 0.3573 ± 0.0006 |
| CBP | **0.9711** ± 0.0002 | **0.9403** ± 0.0003 | **0.9925** ± 0.0002 | **0.9944** ± 0.0003 | **0.9478** ± 0.0003 | **0.8875** ± 0.0004 | **0.9678** ± 0.0003 | **0.9906** ± 0.0002 | **0.9116** ± 0.0002 |

in Appendix A. We propose three lemmas (Lemma 1-3) that establish the convergence of CBP in Appendix B and provide a detailed analysis of its computational complexity in Appendix C.

## 6 EXPERIMENTS

① **Dataset and Preprocessing:** We evaluated our method on both synthetic and real-world fMRI datasets. **i) Synthetic datasets** include: (1) Three-task switching scenario: a simple setting simulating three task conditions. (2) Ten-task switching scenario: a more challenging setting with ten tasks. **ii) Real fMRI datasets** include: (1) the *Over 100 Task* (OT) dataset Nakai & Nishimoto (2020a;b), covering 103 cognitive tasks; and (2) the *HCP dataset* Van Essen et al. (2012), which contains seven categories of cognitive tasks, each of which has multiple subtasks. (3) the *Think Like an Expert* (TLAE) dataset Meshulam et al. (2020; 2021), a longitudinal study with 20 students and 5 experts scanned over six sessions (five lectures and one review/exam session). See the appendix D for more details.

② **Evaluation of CBP: i) Comparison with Baselines:** To systematically evaluate the effectiveness of the proposed method, we compared CBP against several baselines, including deep learning models (NetFormer Lu et al. (2025), LtrRNN Pellegrino et al. (2023), AMAG Li et al. (2023), HGFM Han et al. (2025), and BrainMoE Wei et al. (2025)) and explicit state-space models (SLDS, rSLDS Linderman et al. (2016; 2017), and mp-rSLDS Glaser et al. (2020)) (details in Appendix E), on both synthetic and real fMRI datasets. To ensure a fair comparison, we trained all baseline architectures from scratch under the exact same data scale and task configuration as CBP, without relying on any large-scale pretraining. **The comparison with deep models is intended to show that even without deep learning architectures, CBP can achieve reconstruction performance comparable to deep networks while providing the interpretability that deep models typically lack**. The implementation details of our model are provided in Appendix F. We use mean squared error (MSE) and Pearson correlation as evaluation metrics. Results in Table 1, Table 8 and Fig. 6 show that: on synthetic data, CBP not only accurately reconstructs the observed signals $Y$ but also precisely recovers the ground-truth cognitive bases $A$, latent trajectories $X$; on real fMRI data, CBP significantly outperforms most baselines. Statistical significance analyses (Wilcoxon signed-rank tests)

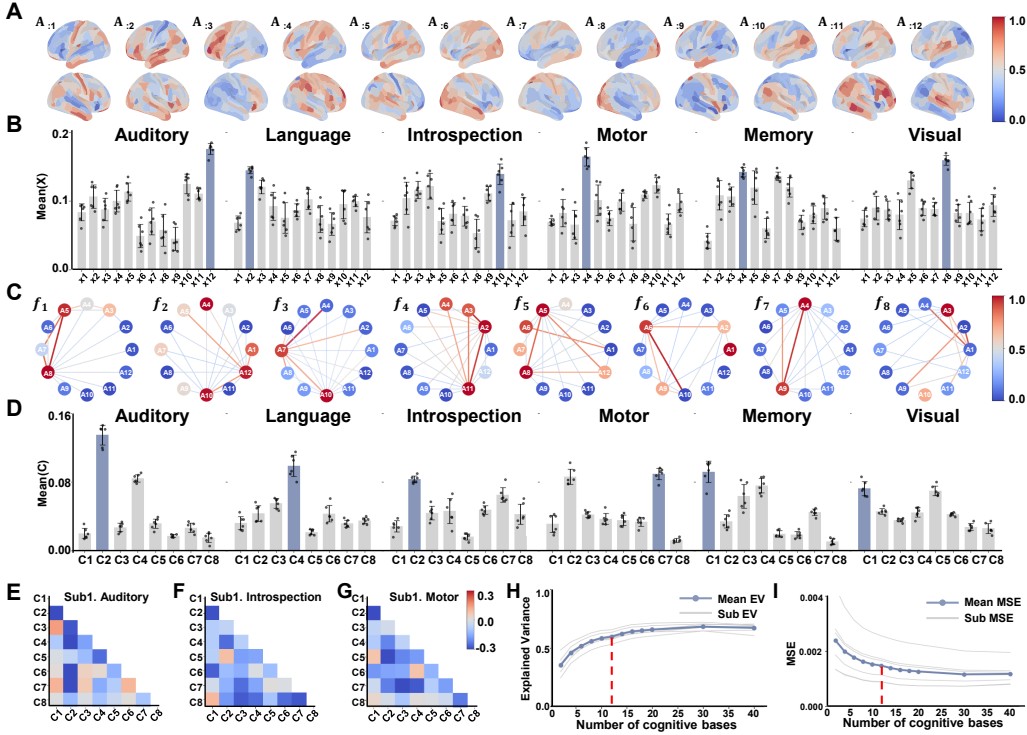

Figure 3: CBP discovers cognitive bases and connectivity patterns from the *Over 100 Task* (OT) dataset. (A) Cortical distribution of $K = 12$ shared cognitive bases from the OT dataset, closely aligned with canonical functional networks. (B) Mean activation (Mean($X$)) of each basis across six task categories, revealing task-specific recruitment. (C) Eight shared connectivity patterns ($\{f_p\}_{p=1}^{8}$); edge color denotes inter-basis connection strength and node color encodes self-connection weight, illustrating diverse coordination motifs. (D) Mean activation (Mean($C$)) of each pattern across task categories, showing pronounced task-specific engagement. (E–G) Correlation matrices of activation coefficients $\{C_p\}_{p=1}^{8}$ under different tasks, showing that distinct tasks recruit different pattern combinations to organize neural dynamics. (H–I) Model selection for $K$: explained variance and reconstruction error versus $K$, with performance saturating beyond $K = 12$ (elbow criterion).

show that CBP outperforms BrainMoE on datasets where the difference is statistically significant, and performs comparably on datasets where the numerical differences are not significant (details in Appendix P). Despite not using deep learning, CBP achieves near–SOTA performance. We attribute this advantage to two key design choices: (1) CBP explicitly models cross-task shared information via multi-cognitive-task learning, yielding more robust modeling under multi-cognitive-task conditions, whereas existing approaches typically focus on single-task settings; and (2) CBP leverages dictionary learning to identify multiple co-active connectivity patterns and cognitive bases, a capability absent in prior methods. **ii) Ablation Study:** We conducted an ablation study to evaluate the contributions of the proposed framework and its key components, using both synthetic data and the OT dataset. We considered five model variants: CBP.a, CBP.b, CBP.c, CBP.d, CBP.e (details in Appendix G). As shown in Table 2 and Fig. 7, CBP consistently outperforms all ablated variants across datasets, confirming the necessity of each component in the proposed design (the qualitative interpretation in Appendix R). These results highlight three key findings: (1) Cross-task learning leverages effectively shared information and improves reconstruction performance; (2) Dictionary learning further enhances the utility of shared information; (3) Explicitly modeling latent dynamics significantly improves temporal modeling capacity.

③ **CBP Discovers Cognitive Basis space and Connectivity Patterns in the 103 Cognitive Task Dataset:** With model performance ensured, we then test the scientific hypothesis that neural dy-

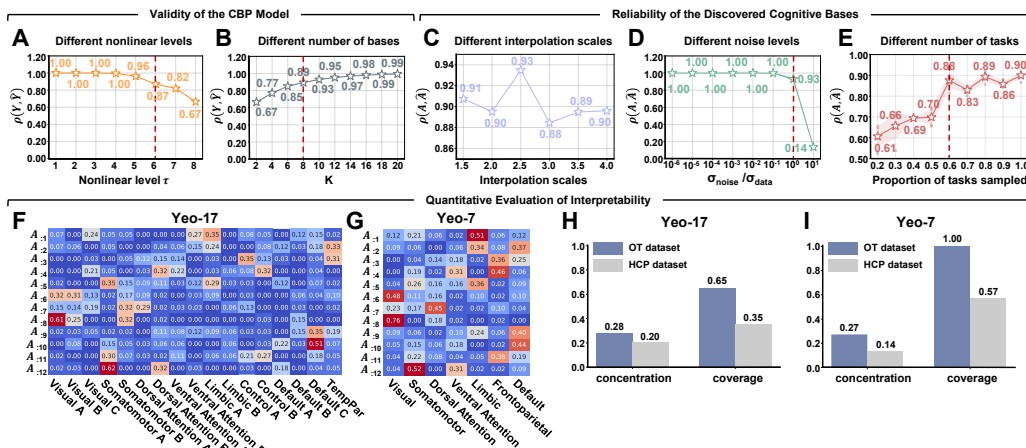

Figure 4: *Validity and Reliability Evaluation:* (A) CBP maintains high reconstruction accuracy across a wide nonlinearity range and remains stable under strong nonlinearity $\tau = 6$. (B) CBP's performance declines under extreme nonlinearity ($\tau = 8$), while larger $K$ improves reconstruction in line with a piecewise-linear approximation. (C) Bases remain highly consistent across different temporal resolutions (upsampling factors 1.5–4.0). (D) Bases remain robust to Gaussian noise even when the noise ratio reaches $\frac{\sigma_{noise}}{\sigma_{data}} = 1$. (E) Bases become stable once the number of sampled tasks reaches a sufficient scale. (F–G) Overlap with Yeo-17/7 networks shows strong correspondence between OT-derived bases and canonical functional networks. (H–I) Bases trained on OT show higher concentration and coverage than bases trained directly on HCP, indicating stronger interpretability.

namics across tasks are governed by a shared set of cognitive bases and connectivity patterns, such that task-evoked activity traces distinct trajectories of a unified cognitive system. Using the over 100 task dataset, we learn $K = 12$ shared cognitive bases (Fig. 3A) whose cortical distributions closely match canonical functional networks, capturing large-scale brain organization. To assess task relevance, we group tasks into six categories (Auditory, Language, Introspection, Motor, Memory, Visual) and compute category-wise mean activations Mean($X$) (Appendix Q, Fig. 3B). The profiles are highly discriminative: in Auditory tasks, $A_{:12}$ is localized to the auditory cortex and shows the strongest activation, reflecting speech processing demands; in Language tasks, $A_{:2}$ is localized to the early visual cortex and is most active, engaging orthographic processing and serving as the visual entry point for language comprehension. Other categories exhibit similar one-to-one correspondences between strongly activated bases and functionally relevant brain regions. Together, these results show that CBP learns biologically meaningful cognitive bases that robustly differentiate task categories and align with canonical networks, providing clear evidence of task-level interpretability.

Next, we test our core hypothesis that not only are the cognitive bases task-general, but their connectivity patterns remain stable and capture coordination among functional modules. We analyze the relationships among the learned bases and identified $P = 8$ shared connectivity patterns (Fig. 3C). Edge colors encode inter-basis connection strengths, and each pattern exhibits distinct topology and strength profiles, revealing diverse functional coordination motifs. Node colors denote self-connection strength, indicating each basis's relative weight within a pattern. To assess whether these patterns participate in task-evoked dynamics, we compute their mean activation (Mean($C$)) across task categories (Appendix Q, Fig. 3D). Results reveal strong task-specific modulation. In Auditory tasks, $f_2$ shows maximal activation, centered on $A_{:12}$ (auditory cortex), forming a perception–action loop with $A_{:5}$ (SMN) for articulatory simulation and rhythmic synchronization, and strongly coupling with $A_{:10}$ (multimodal association areas) to integrate auditory, visual, and semantic information for speech comprehension. In Language tasks, $f_4$ is most strongly activated, coordinating $A_{:2}$ (early visual cortex), $A_{:3}$ (semantic/lexical regions), and $A_{:11}$ (DMN) to jointly support orthographic encoding, semantic processing, and memory retrieval. Other categories show similar correspondences, with highly activated patterns consistently matching the cognitive demands of each domain. Together, these results demonstrate that the learned connectivity patterns are struc-

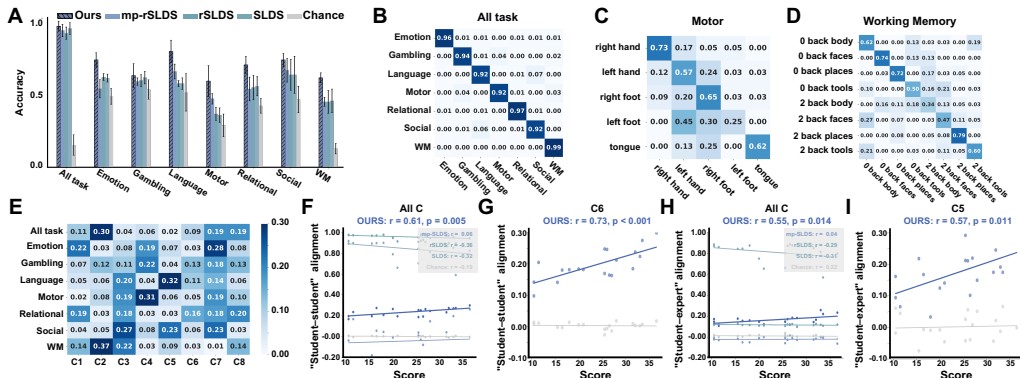

Figure 5: CBP results on task decoding and learning outcome prediction. (A) Task classification accuracy on HCP: CBP significantly outperforms the baselines. (B–D) Confusion matrices across categories. (E) Feature-importance heatmap of $\{C_p\}_{p=1}^P$. (F–G) "Student–student" alignment during learning, aggregated across all patterns, positively correlates with learning outcome ($r = 0.61, p = 0.005$) and outperforms the baselines; $f_6$ ($C_6$) shows the strongest effect ($r = 0.73, p < 0.001$), indicating consistent recruitment of the salience–integration pathway by high performers. (H–I) "Student–expert" alignment during review, aggregated across all patterns, similarly predicts performance ($r = 0.55, p = 0.014$) and outperforms the baselines; $f_5$ ($C_5$) is most predictive ($r = 0.57, p = 0.011$), revealing a multimodal coordination pathway underlying convergence toward expert-like strategies.

turally stable and systematically differentiate task-specific coordination requirements, supporting our hypothesis.

Fig. 3E–G and Fig. 8 visualize the coefficient ($C_p = \{c_{m,t}^{p,(s)}\}_{t=1}^{T_m^s}$) correlation matrix for each task and subject, showing that tasks engage distinct and co-activated connectivity patterns to organize neural dynamics, supporting our hypothesis that these patterns encode task-specific coordination demands. Finally, following the work MacDowell & Buschman (2020), we systematically evaluate the selection of the number of cognitive bases $K$ and connectivity patterns $P$. As shown in Fig. 3H-I, increasing $K$ beyond 12 yields only marginal performance gains. Based on a joint consideration of explained variance, reconstruction error, and model complexity (i.e., the elbow criterion), we fix $K = 12$. We then apply the PhenoGraph algorithm to the learned connectivity patterns for unsupervised clustering Levine et al. (2015); Nicosia et al. (2009), thus producing eight stable clusters, leading us to set $P = 8$ to ensure that the resulting patterns are both robust and biologically interpretable.

④ **Validity and Reliability Evaluation:** We further evaluate the validity of the model using a simulation system with tunable nonlinearity (see Appendix M). CBP maintains high reconstruction accuracy across a wide nonlinearity range and remains stable even under strong nonlinearity ($\tau = 6$) (Fig. 4A). Its performance declines under extreme nonlinear conditions ($\tau = 8$), while increasing the number of bases $K$ improves reconstruction accuracy in a manner consistent with a piecewise-linear approximation (Fig. 4B). We then evaluate the reliability of the cognitive bases: (i) When upsampling the time axis with interpolation factors from 1.5 to 4.0 to simulate higher temporal resolution (shorter TR), the learned bases remain highly consistent with the originals (Fig. 4C). (ii) When progressively adding Gaussian noise to the fMRI signals, the learned bases remain highly similar to the originals even when the noise ratio reaches $\frac{\sigma_{noise}}{\sigma_{data}} = 1$ (Fig. 4D). (iii) When randomly sampling different numbers of cognitive tasks from the OT dataset and learning the bases independently, the results show that once task sampling reaches a sufficient scale, the learned bases become stable (Fig. 4E). Finally, we conduct quantitative evaluations of interpretability (the definitions of metrics in Appendix N). (i) Computing the overlap between each basis and the Yeo-17 or Yeo-7 functional networks yields a basis–network mapping matrix, which reveals strong correspondence between OT-derived bases and canonical functional networks (Fig. 4F–G). (ii) Comparing bases trained directly on HCP with those trained on OT shows that $A_{OT}$ exhibits significantly higher con-

centration and coverage than $A_{HCP}$ (Fig. 4H–I), indicating stronger interpretability. This difference likely reflects the limited number of tasks and uneven task durations in HCP, which make it difficult to learn the stable and interpretable bases.

⑤ **CBP Reveals Interpretable Cognitive Bases and Connectivity Patterns for Predicting Task-Related Variables:** We conduct systematic evaluations of CBP on the HCP and multi-stage learning (TLAE) datasets to validate its capability of inferring task-related variables and uncovering cognitive mechanisms. We first learn a shared cognitive basis $A$ and connectivity patterns $\{f_p\}_{p=1}^P$ on the OT dataset and directly generalize them to HCP and TLAE, achieving strong reconstruction performance (Table 1), demonstrating robust cross-dataset generalization.

On HCP, we train a logistic regression classifier using pattern activations $C_p = \{c_t^p\}_{t=1}^T$ as features. CBP accurately predicts seven major task categories and further discriminates fine-grained tasks within categories, significantly outperforming the baselines (Fig. 5A). The confusion matrix (Fig. 5B-D) and feature importance map (Fig. 5E) quantify performance and reveal contributions of individual patterns. $C_2$ is most informative, with interaction pattern $f_2$ anchored in auditory cortex ($A_{:12}$) and connecting multimodal integration area ($A_{:10}$) and motor loop ($A_{:5}$), highlighting systematic differences in auditory reliance and integration across tasks. In emotion tasks, $C_7$ contributes most, with $f_7$ centered on fear-sensitive $A_{:9}$ and linking $A_{:4}$ (FPN) and $A_{:5}$ (SMN), capturing differences in perception, attention, and motor preparation. Fig. 5A–E illustrate task decoding analysis for subject01. We further evaluate task decoding performance for all subjects (Fig. 9 and Fig. 10).

On the TLAE dataset, we evaluate CBP's ability to predict learning performance. From a group-reference perspective, we compute the "student–student" pattern activation alignment, defined as the similarity between each student's activation coefficients $C_p$ and the population mean across the learning phase. Higher alignment is associated with significantly better learning outcomes. Compared with the baselines, CBP provides more accurate predictions of learning outcomes (Fig. 5F). Importantly, the strength of the correlation varies across patterns, with $f_6$ showing the strongest association (Fig. 5G and Fig. 11A). $f_6$ is anchored in $A_{:6}$ (VAN) and connects early visual cortex ($A_{:2}$), temporoparietal association area ($A_{:9}$), and multimodal integration area ($A_{:10}$), forming a bottom-up salience detection and multimodal integration circuit. This suggests that high-performing students consistently recruit this pathway, facilitating efficient information integration and contextual processing. Finally, from an expert-reference perspective, we compute the "student–expert" alignment by comparing each student's activation patterns during the review phase with the expert population mean. This metric also robustly predicts learning performance. CBP consistently outperforms the baselines in predicting learning outcomes (Fig. 5H). Among all patterns, $f_5$ exhibits the strongest correlation with performance (Fig. 5I and Fig. 11B). $f_5$ connects $A_{:5}$ (SMN), $A_{:8}$ (VIS), and $A_{:12}$ (auditory cortex), delineating a multimodal circuit spanning visual, auditory, and sensorimotor networks. This pathway likely reflects the convergence of students' representations toward expert-like strategies during review, a process tightly linked to performance improvement.

## 7 DISCUSSION

Understanding the brain dynamics that support human cognition requires models that can deliver both strong predictive capacity and interpretability across tasks. Yet current solutions either use deep learning models that produce highly entangled and difficult-to-interpret representations, or employ explicit state-space models that, despite being interpretable, are limited to single-task settings. To address this limitation, we present CBP, an interpretable, non–deep-learning framework designed for cross-cognitive-task brain modeling. CBP captures the shared information across diverse tasks, enabling it to recover latent components with high fidelity while achieving performance comparable to state-of-the-art deep models. Notably, CBP consistently identifies a robust set of cognitive bases and connectivity patterns in the human brain. Their stability is corroborated through comprehensive quantitative analyses and a series of perturbation experiments. Building on these shared cognitive components, CBP generalizes effectively to both the HCP and multi-stage learning datasets, where it reliably predicts task states and learning outcomes and uncovers the key connectivity patterns that drive these processes. Collectively, these results position CBP as an interpretable and scalable framework for modeling large-scale cognitive dynamics, providing mechanistic insights into how cognition is organized across tasks. Limitations and future work are provided in Appendix L.

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

## A  THE ALGORITHM OF CBP

We summarize the algorithm of CBP as follows:

## B  CONVERGENCE ANALYSIS

In this section, we analyze the convergence of the CBP using the Alternating Convex Search (ACS) algorithm, which solves a convex optimization problem with multiple regularization terms. Based on the work Gorski et al. (2007); Liu et al. (2023); Zheng et al. (2020), we need the following theorems:

**Theorem 1** *Let $\mathcal{T} \in \mathbb{R}^{r \times s}$, and we assume that $f : \mathcal{T} \to \mathbb{R}$ is bounded. If the optimization problem for each variable in every iteration is solvable, then the sequence $\{f(w_i)\}$ (where $w_i \in \mathcal{T}$) produced by the ACS algorithm will exhibit monotonic convergence.*

**Theorem 2** *Let $\mathcal{U} \subset \mathbb{R}^r$ and $\mathcal{V} \subset \mathbb{R}^s$ be closed sets, and let $f : \mathcal{U} \times \mathcal{V} \to \mathbb{R}$ be a continuous function. If the optimization problem for each variable can be solved at every iteration, it holds:*

---

**Algorithm 1** CBP training

---

1: **Input:** $P, K, \lambda_a, \lambda_x, \lambda_c, \lambda_\rho, \alpha_1, \alpha_2, \alpha_3$
2: **Initialize:** $A, X_m^{(s)}, c_{m,t}^{p,(s)}, \{f_p\}_{p=1}^P$
3: **repeat**
4:     **for** each task $m$ **do**
5:         Update $A$ according to equation 2
6:         Update $X_m^{(s)}, c_{m,t}^{p,(s)}$ according to equation 3
7:         Update $\{f_p\}_{p=1}^P$ according to equation 4
8:     **end for**
9: **until** convergence
10: **Output:** $A, X_m^{(s)}, c_{m,t}^{p,(s)}, \{f_p\}_{p=1}^P$

---

    *1. If the sequence $\{w_i\}$ (where $w_i \in \mathcal{T}$) produced by the ACS algorithm is contained within a compact set, then the number of accumulation points of the sequence will be at least one.*

    *2. For each accumulation point $w^*$ of the sequence $\{w_i\}$ (where $w_i \in \mathcal{T}$), we have:*

        *(a) If fixing all other variables results in a unique optimal solution for one variable, then all accumulation points corresponding to that variable will have the same function value and will be local optimal.*

        *(b) If there exists a unique optimal solution for one variable, then $\lim_{i \to \infty} \|w_{i+1} - w_i\| = 0$, and the accumulation points will form a compact set $\mathcal{S}$.*

According to Theorem 1 and Theorem 2, we present the following lemmas.

**Lemma 1** *The sequence $\{A, X_m^{(s)}, c_{m,t}^{p,(s)}, f_p\}$ generated by the CBP algorithm is monotonic and convergent in objective value.*

**Proof.** We apply the ACS algorithm to solve the optimization problem. Specifically, in Equation 2, we fix $X_m^{(s)}, c_{m,t}^{p,(s)}$ and $f_p$, and optimize $A$. The problem for $A$ is clearly strongly convex, as it involves a quadratic loss with respect to $A$, ensuring uniqueness and monotonic convergence of the solution. Next, we fix $A$ and $f_p$ to optimize $X_m^{(s)}$ and $c_{m,t}^{p,(s)}$ as in Equation 3, and then fix $X_m^{(s)}$, $c_{m,t}^{p,(s)}$ and $A$ to optimize $f_p$ as in Equation 4, both of which are convex, ensuring monotonic convergence. By Theorem 1, since all optimization steps involve convex subproblems that are solvable, the sequence $\{A, X_m^{(s)}, c_{m,t}^{p,(s)}, f_p\}$ generated by the algorithm converges monotonically.

**Lemma 2** *The sequence $\{A, X_m^{(s)}, c_{m,t}^{p,(s)}, f_p\}$ generated by the algorithm has at least one accumulation point. All such accumulation points correspond to local optima of the objective function and attain the same function value.*

**Proof.** Since the sequence $\{A, X_m^{(s)}, c_{m,t}^{p,(s)}, f_p\}$ is bounded due to the coercive regularization terms, it lies in a compact set. Thus, by Theorem 2.1, there is at least one accumulation point. Each subproblem involves a strictly convex objective (as shown in Lemma 1), ensuring that the solution for each variable is unique. According to Theorem 2.2, every accumulation point corresponds to a local optimum. Moreover, since the objective function is monotonically decreasing and bounded below, all accumulation points must yield the same function value.

**Lemma 3** *The sequence $\{A, X_m^{(s)}, c_{m,t}^{p,(s)}, f_p\}$ satisfies the following convergence condition:*

$$\lim_{t^* \to \infty} \left( \|A^{t^*+1} - A^{t^*}\| + \|X_m^{(s)^{t^*+1}} - X_m^{(s)^{t^*}}\| + \|c_{m,t}^{p,(s)^{t^*+1}} - c_{m,t}^{p,(s)^{t^*}}\| + \|f_p^{t^*+1} - f_p^{t^*}\| \right) = 0,$$

*where $t^*$ is the number of iterations.*

Lemma 1 proves the convergence of the optimization problem, while Lemma 2 and Lemma 3 detail the process for obtaining the optimal values. Together, they ensure that the sequence converges to the desired solution.

## C  TIME COMPLEXITY CALCULATION

The optimization of our model parameters relies on the alternating updates of three main procedures in each iteration:

**i) Cognitive Bases Update**: We concatenate the data from $s \in [1, S]$ subjects and $m \in [1, M]$ tasks along the time dimension to construct the data matrix $Y \in \mathbb{R}^{N \times T_{total}}$, where $T_{total} = \sum_{s=1}^{S} \sum_{m=1}^{M} T_m^s$. As detailed in Equation 2, the time complexity of this operation is primarily determined by two main components: $||Y - AX||^2$ and $\lambda \cdot tr(A^T LA)$. The time complexity of this operation can be calculated as: $O(N \cdot K \cdot T_{total}) + O(N^2 K) = O(N \cdot K \cdot T_{total})$.

**ii) Latent State and Dynamic Coefficients Update**: When updating the latent state, $X$, the time complexity arises primarily from $||Y - AX||^2$ and $\sum_t ||x_{m,t}^{(s)} - \sum_{p=1}^{P} f_p c_{m,t}^{p,(s)} x_{m,t-1}^{(s)}||^2$, resulting in $O(T_{total} \cdot (N \cdot K + P \cdot K^2))$. The time complexity to update the dynamic coefficients, $C = \{c_{m,t}^{p,(s)}\}_{t=1}^{T_{total}}$, is derived from the calculation of the $\sum_t ||x_{m,t}^{(s)} - \sum_{p=1}^{P} f_p c_{m,t}^{p,(s)} x_{m,t-1}^{(s)}||^2$, which is $O(T_{total} \cdot P \cdot K^2)$, as detailed in Equation 3.

**iii) Connectivity Patterns Update**: According to the Equation 4, this step is responsible for identifying the connectivity patterns $f_p$. The time complexity of this update process lies mainly in the $\sum_t ||x_{m,t}^{(s)} - \sum_{p=1}^{P} f_p c_{m,t}^{p,(s)} x_{m,t-1}^{(s)}||^2$ and the $\lambda_\rho \sum_{p_1, p_2, (p_1 \neq p_2)} \rho(f_{p_1}, f_{p_2})$. The total time complexity is $O(T_{total} \cdot P \cdot K^2 + P^2 \cdot K^2) = O(T_{total} \cdot P \cdot K^2)$.

Combining these terms, the per-iteration complexity is $O(T_{\text{total}}(N \cdot K + P \cdot K^2))$, which is linear in the total number of time points $T_{\text{total}}$ (and thus linear in the number of tasks and subjects). Since $K$ and $P$ are small in practice (e.g., $K = 12$, $P = 8$) and $N = 360$ is fixed by the atlas, the overall training cost scales well to larger multi-task datasets. All updates rely on vectorized matrix operations, making CBP efficient on CPU and naturally compatible with GPU acceleration. In our experiments, training on the largest dataset fits comfortably on a single workstation; no additional tricks such as low-rank approximations are required.

During the three procedural steps, the **cognitive bases update** and **connectivity patterns update** constitute the primary source of computational overhead. In contrast, within the **generalization scenario** (Think Like an Expert (TLAE) dataset), the model parameters $A$ (cognitive bases) and $f_p$ (connectivity patterns) have been trained and fixed, substantially accelerating the process. The main task of the model is to **infer** the corresponding **latent states** $X$ and **dynamic coefficients** $C$ for new observed data, and these two processes are highly efficient. In this process, the core calculation is the inference of parameters $X_{new}$ and $C_{new}$ for new observed data $Y_{new}$ through alternating optimization. During the update of the latent states $X$, the time complexity primarily stems from the $||Y - AX||_2^2$ and the $\sum_t ||x_{m,t}^{(s)} - \sum_{p=1}^{P} f_p c_{m,t}^{p,(s)} x_{m,t-1}^{(s)}||_2^2$, resulting in a complexity of $O(T_{total} \cdot (N \cdot K + P \cdot K^2))$. Similarly, the time complexity for updating the dynamic coefficients $C$ is dominated by the $\sum_t ||x_{m,t}^{(s)} - \sum_{p=1}^{P} f_p c_{m,t}^{p,(s)} x_{m,t-1}^{(s)}||_2^2$, which is $O(T_{total} \cdot P \cdot K^2)$.

## D  DETAILS OF DATASETS

### D.1  SYNTHETIC DATASET

**Three-Task Synthetic Dataset:**

To test our core hypothesis that neural dynamics across cognitive tasks are jointly governed by a set of shared cognitive bases and their connectivity patterns, we conceptualize task-evoked brain activity as distinct trajectories of a unified cognitive system, each reflecting a task-specific unfolding of these shared components. We first generate a synthetic dataset consisting of three tasks ($m = 3$). All tasks share a common cognitive basis matrix $A \in \mathbb{R}^{N \times K}$, where $K = 3$ denotes the number of cognitive bases and $N = 21$ the number of brain regions, with each column representing a cognitive basis across brain regions.

Tasks also share a set of connectivity patterns $\{f_p\}_{p=1}^{P}$, where each $f_p \in \mathbb{R}^{K \times K}$ characterizes the connectivity among cognitive bases, and $P = 3$ specifies the number of patterns. The coefficient $c_{m,t}^p$ controls the activation strength of the $p$-th connectivity pattern at time $t$ within task $m$. For

simplicity, we assume that each task is dominated by a single connectivity pattern $f_p$ throughout its duration, resulting in three distinct task types. Task durations are randomly sampled from the interval $[10, 30]$ time points, and their order is randomized. To ensure smooth transitions between tasks, we convolve $c_{m,t}^p$ with a Gaussian kernel (kernel size 6, standard deviation 0.7), yielding a gradual switching profile. The total time series length is set to $T = 500$.

Given $c_{m,t}^p$ and $f_p$, we compute the latent states: $x_{m,t} = \sum_{p=1}^{P} f_p \, c_{m,t}^p \, x_{m,t-1}$, and reconstruct the observed signal as $y_{m,t} = A x_{m,t}$. To approximate measurement noise observed in empirical data, we add Gaussian noise with a standard deviation of 0.0001 to $y_{m,t}$, producing the final observed signal $Y$, while retaining its underlying components $A, f_p, c_{m,t}^p$ and $x_{m,t}$ for subsequent evaluation.

**Ten-Task Synthetic Dataset:**

To evaluate model performance under more challenging scenarios, we extend the experiment to a setting with ten cognitive tasks ($m = 10$). Unlike the simplified assumption that each task is dominated by a single connectivity pattern, we model each task as a convex combination of three connectivity patterns $\{f_p\}_{p=1}^3$. The task $m$ at time $t$ is defined as a convex combination of the three patterns $\sum_{p=1}^3 c_{m,t}^p f_p$, $c_{m,t}^p \geq 0$, $\sum_{p=1}^3 c_{m,t}^p = 1$, where $c_{m,t}^p$ denotes the relative contribution of pattern $p$ at time $t$. To generate diverse task compositions, we uniformly sample ten weight vectors $\{c_m^p\}_{m=1}^{10}$ from a Dirichlet distribution with parameter $\alpha = (1,1,1)$, yielding ten distinct tasks. All other experimental settings are kept identical to the previous experiments.

## D.2 OVER 100 TASK DATASET

We used the publicly available Over 100 Task fMRI Dataset. This dataset records whole-brain activity from six healthy participants while they performed 103 diverse cognitive tasks. These tasks cover a wide range of cognitive domains, including perception, motor control, language processing, memory, and executive function. They are grouped into six major categories: Auditory, Language, Introspection, Motor, Memory, and Visual. This design provides a systematic sampling of the brain's functional organization across multiple cognitive states. For each participant, the dataset includes high-resolution T1-weighted structural images and functional MRI signals acquired with a multi-band echo-planar imaging (EPI) sequence. This acquisition protocol provides both high temporal and spatial resolution. Each participant completed 18 runs in total, covering all 103 tasks. The dataset also provides event files specifying stimulus onset times, durations, and task labels, which enable precise alignment between task events and neural signals. To ensure standardized and reproducible analysis, we preprocessed the raw fMRI data using the fMRIPrep pipeline. The preprocessing steps consisted of slice timing correction, motion correction, spatial normalization, and nuisance regression. After preprocessing, we parcellated the whole brain into 360 regions using the HCP multimodal parcellation (MMP) atlas and standardized the BOLD signal within each region. The resulting quality-controlled time series were then used for subsequent modeling and analysis.

## D.3 HCP DATASET

The **Human Connectome Project (HCP)** is a large-scale neuroimaging initiative that provides high-resolution structural and functional MRI data from healthy adult participants. The dataset includes task-based fMRI recordings. The task-based fMRI data covers **seven categories of cognitive tasks**, each designed to elicit specific neural responses. The seven categories of cognitive tasks are described as follows.

**Emotion**:

fear: Participants are shown images of fearful faces and are asked to match them based on emotional expression.

neutral (neut): Participants are shown images of neutral faces and are tasked with matching these images to others with similar expressions.

**Gambling**:

win: Participants engage in a gambling task where they are presented with scenarios that result in monetary rewards, and their responses are monitored.

loss: Participants are presented with scenarios involving losses and are asked to respond accordingly to the outcomes.

**Language**:

story: Participants listen to or read stories and are required to answer questions or make judgments based on the story content.

math: Participants perform mathematical tasks, such as solving problems or performing calculations, assessing numerical reasoning.

**Motor**:

right hand (rh): Participants are asked to perform movements with their right hand, such as squeezing a ball or pressing a button.

left hand (lh): Similar to the right hand task, but with movements executed using the left hand.

right foot (rf): Participants perform movements with their right foot.

left foot (lf): Participants perform movements with their left foot.

tongue (t): Participants are asked to perform tongue movements, such as sticking out the tongue or moving it within the mouth.

**Relational**:

relation: Participants are presented with pairs or groups of items and are tasked with identifying relationships between them, requiring reasoning and comparison.

match: In this task, participants are asked to determine whether pairs of items match based on specific criteria, such as shape or category.

**Social**:

mental (mentalizing): Participants are presented with scenarios involving others' mental states (e.g., beliefs or intentions) and are asked to infer what the other person might be thinking.

random (rnd): This task involves random or control stimuli, helping to isolate the brain regions involved in social cognition.

**Working Memory**:

0-back (body, faces, places, tools): In the 0-back task, participants see a series of stimuli (e.g., body images, faces, places, or tools) and are required to respond when a stimulus matches the one presented at the start.

2-back (body, faces, places, tools): In the 2-back task, participants are asked to respond when a stimulus matches the one presented two steps earlier, requiring greater working memory capacity.

We randomly selected fMRI signals from 10 subjects across 7 major cognitive tasks and processed them using the HCP minimal preprocessing pipeline. The brain was parcellated into 360 regions using the MMP360 atlas, and the signals from each region were standardized to enable consistent downstream analysis.

### D.4  THINK LIKE AN EXPERT DATASET

We used the publicly available Think Like an Expert fMRI dataset. This dataset includes fMRI data from 20 "student" participants and 5 "expert" participants. The student participants were enrolled in an introductory computer science course and underwent six fMRI scanning sessions over the duration of the course. During the first five sessions, students watched course lecture videos inside the scanner. In the final session, they viewed review videos from the previous weeks and completed a final exam, which required verbal responses to visually presented questions. The expert participants only took part in the final session, serving as a reference group. A T1-weighted structural image was acquired during each session. The dataset also provides exam scores for both students and experts, enabling quantitative assessment of learning performance and comprehension. The experimental paradigm covers the full course curriculum, including fundamental topics such as conditional state-

ments and loops, functions and libraries, abstract data types, and computational theory. To ensure standardized and reproducible analysis, we preprocessed the raw fMRI data using the fMRIPrep pipeline, including slice-timing correction, motion correction, spatial normalization, and nuisance regression. After preprocessing, we parcellated the whole brain into 360 regions using the HCP multimodal parcellation (MMP) atlas and standardized the BOLD signals within each region. The resulting quality-controlled time series were used for subsequent modeling and analyses.

# E    DETAILS OF BASELINE

**ltrRNN** Pellegrino et al. (2023): A recurrent network with low-tensor-rank weights, modeling neural population dynamics and capturing low-dimensional structure evolving during learning.

**AMAG** Li et al. (2023): A graph neural network with additive and multiplicative message passing to forecast neural activity and recover interactions.

**NetFormer** Lu et al. (2025): A transformer-based model that uses state-dependent attention on neuron activity histories to capture nonlinear, nonstationary connectivity.

**HGFM** Han et al. (2025): A hypergraph neural foundation model that encodes high-order dependencies among multiple ROIs to provide generalizable representations of large-scale neural dynamics.

**BrainMoE** Wei et al. (2025): A mixture-of-experts brain foundation model that integrates cognition-specific experts through a cross-attention cognition adapter to learn robust, state-aware representations of large-scale neural dynamics.

**SLDS** Linderman et al. (2020): A probabilistic model that segments time series into discrete latent modes, each governed by a linear dynamical system.

**rSLDS** Linderman et al. (2016; 2017): An extension of SLDS that models mode transitions as functions of observations or continuous latent states, capturing flexible switching behavior.

**mp-rSLDS** Glaser et al. (2020): A multi-population extension of rSLDS where each population has its own latent variables and interacts through a shared low-dimensional space, identifying populations driving transitions.

# F    IMPLEMENTATION DETAILS

The full set of parameters for the experiments conducted on the Three-Task synthetic dataset is provided in Table 3, on Ten-Task synthetic dataset in Table 4, on the OT dataset in Table 5, on the HCP dataset in Table 6, and on the TLAE dataset in Table 7.

Table 3: Parameter values for Three-Task synthetic dataset experiments

| Symbol | Values | Description |
|---|---|---|
| $lr$ | 0.01-0.001 | Learning rate for parameter updates |
| $E$ | 200 | Number of epochs for parameter updates |
| $N$ | 21 | The number of brain regions |
| $T$ | 500 | The number of time points |
| $K$ | 3 | The number of cognitive bases |
| $P$ | 3 | The number of connection patterns |
| $\lambda_a$ | 0.001 | Controls the manifold regularization |
| $\lambda_x$ | 0.001 | Encourages accurate reconstruction of latent dynamics |
| $\lambda_c$ | 0.08 | Regulates the temporal smoothness of the dynamic coefficients |
| $\lambda_\rho$ | 0.001 | The weight of the decorrelation term |
| $\alpha_1$ | 0.005 | Controls the sparsity of dynamic Coefficients |
| $\alpha_2$ | 0.001 | Promotes sparsity within connection patterns |
| $\alpha_3$ | 0.005 | Controls the sparsity of the cognitive bases |

Table 4: Parameter values for Ten-Task synthetic dataset experiments

| Symbol | Values | Description |
|---|---|---|
| $lr$ | 0.01-0.001 | Learning rate for parameter updates |
| $E$ | 200 | Number of epochs for parameter updates |
| $N$ | 21 | The number of brain regions |
| $T$ | 500 | The number of time points |
| $K$ | 3 | The number of cognitive bases |
| $P$ | 3 | The number of connection patterns |
| $\lambda_a$ | 0.001 | Controls the manifold regularization |
| $\lambda_x$ | 0.001 | Encourages accurate reconstruction of latent dynamics |
| $\lambda_c$ | 0.03 | Regulates the temporal smoothness of the dynamic coefficients |
| $\lambda_\rho$ | 0.0005 | The weight of the decorrelation term |
| $\alpha_1$ | 0.01 | Controls the sparsity of dynamic Coefficients |
| $\alpha_2$ | 0.0002 | Promotes sparsity within connection patterns |
| $\alpha_3$ | 0.005 | Controls the sparsity of the cognitive bases |

Table 5: Parameter values for OT dataset experiments

| Symbol | Values | Description |
|---|---|---|
| $lr$ | 0.001 | Learning rate for parameter updates |
| $E$ | 200 | Number of epochs for parameter updates |
| $N$ | 360 | The number of brain regions |
| $T$ | 5058 | The number of time points |
| $K$ | 12 | The number of cognitive bases |
| $P$ | 8 | The number of connection patterns |
| $\lambda_a$ | 0.001 | Controls the manifold regularization |
| $\lambda_x$ | 0.001 | Encourages accurate reconstruction of latent dynamics |
| $\lambda_c$ | 0.005 | Regulates the temporal smoothness of the dynamic coefficients |
| $\lambda_\rho$ | 0.001 | The weight of the decorrelation term |
| $\alpha_1$ | 0.001 | Controls the sparsity of dynamic Coefficients |
| $\alpha_2$ | 0.001 | Promotes sparsity within connection patterns |
| $\alpha_3$ | 0.001 | Controls the sparsity of the cognitive bases |

Table 6: Parameter values for HCP dataset experiments

| Symbol | Values | Description |
|---|---|---|
| $lr$ | 0.001 | Learning rate for parameter updates |
| $E$ | 200 | Number of epochs for parameter updates |
| $N$ | 360 | The number of brain regions |
| $T$ | 1940 | The number of time points |
| $K$ | 12 | The number of cognitive bases |
| $P$ | 8 | The number of connection patterns |
| $\lambda_a$ | / | Controls the manifold regularization |
| $\lambda_x$ | 0.001 | Encourages accurate reconstruction of latent dynamics |
| $\lambda_c$ | 0.001 | Regulates the temporal smoothness of the dynamic coefficients |
| $\lambda_\rho$ | / | The weight of the decorrelation term |
| $\alpha_1$ | 0.001 | Controls the sparsity of dynamic Coefficients |
| $\alpha_2$ | / | Promotes sparsity within connection patterns |
| $\alpha_3$ | / | Controls the sparsity of the cognitive bases |

Table 7: Parameter values for TLAE dataset experiments

| Symbol | Values | Description |
|---|---|---|
| $lr$ | 0.001 | Learning rate for parameter updates |
| $E$ | 200 | Number of epochs for parameter updates |
| $N$ | 360 | The number of brain regions |
| $T$ | 6382 | The number of time points |
| $K$ | 12 | The number of cognitive bases |
| $P$ | 8 | The number of connection patterns |
| $\lambda_a$ | / | Controls the manifold regularization |
| $\lambda_x$ | 0.001 | Encourages accurate reconstruction of latent dynamics |
| $\lambda_c$ | 0.001 | Regulates the temporal smoothness of the dynamic coefficients |
| $\lambda_\rho$ | / | The weight of the decorrelation term |
| $\alpha_1$ | 0.001 | Controls the sparsity of dynamic Coefficients |
| $\alpha_2$ | / | Promotes sparsity within connection patterns |
| $\alpha_3$ | / | Controls the sparsity of the cognitive bases |

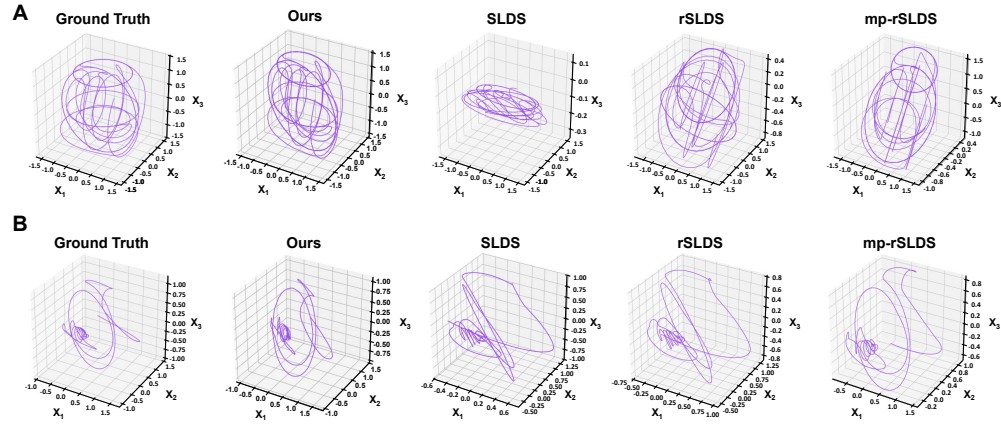

Figure 6: Comparison of our model and baselines in identifying latent trajectories $X$ on Three-Task Synthetic Dataset (A) and Ten-Task Synthetic Dataset (B).

## G  DETAILS OF MODEL VARIANTS FOR ABLATION STUDY

We considered five model variants: **CBP.a** trains on single-task data only and does not learn shared cognitive bases $A$ and connectivity patterns $\{f_p\}_{p=1}^P$ across tasks. **CBP.b** removes the similarity regularization on cognitive bases. **CBP.c** removes the sparsity regularization on cognitive bases. **CBP.d** removes the latent dynamics module. **CBP.e** replaces the dictionary-learning step with PCA.

## H  SUPPLEMENTARY RESULTS OF CBP ON SYNTHETIC DATASETS

We present visual comparisons of latent trajectory $X$ identification by our model and baselines on Three-Task Synthetic Dataset (Fig. 6A) and Ten-Task Synthetic Dataset (Fig. 6B). We present visual comparisons of latent trajectory $X$ identification between our model and its ablation variants on Three-Task Synthetic Dataset (Fig. 7A) and Ten-Task Synthetic Dataset (Fig. 7B).

## I  SUPPLEMENTARY INTERPRETABILITY RESULTS OF CBP ON OT

We provide the coefficient ($C_p = \{c_{m,t}^{p,(s)}\}_{t=1}^{T_m^s}$) correlation matrix for each task and subject (Fig. 8).

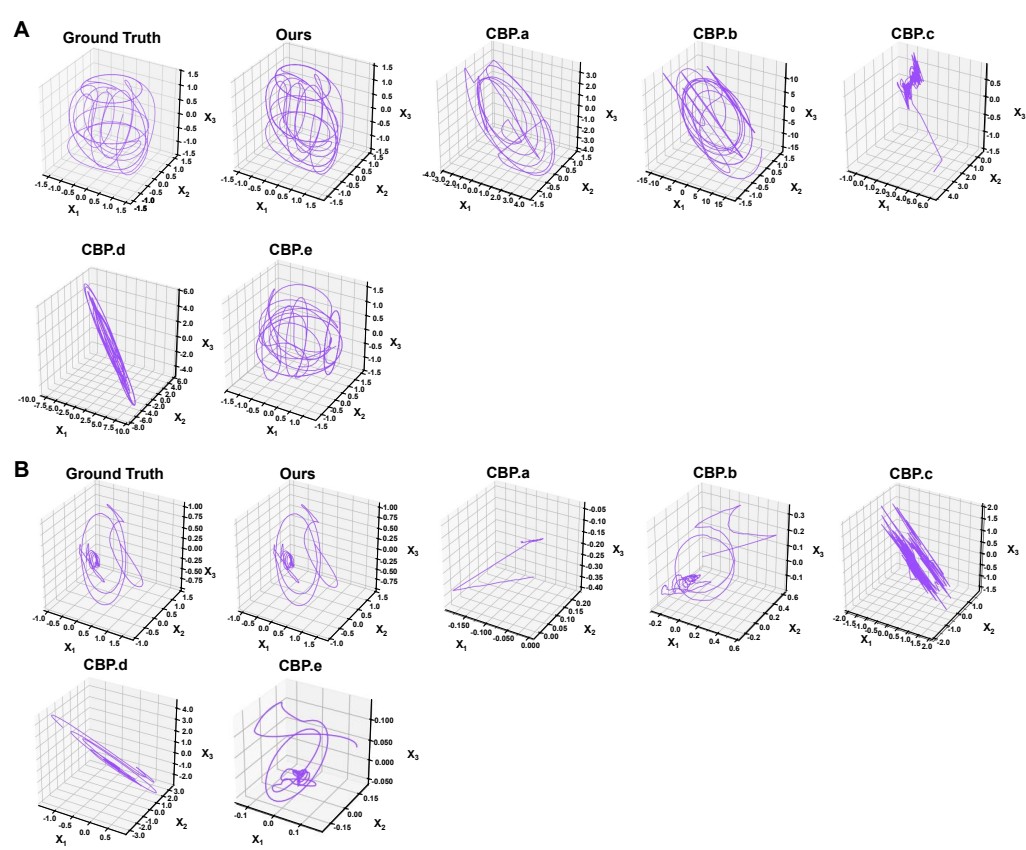

Figure 7: Comparison of our model and its ablation variants in identifying latent trajectories $X$ on Three-Task Synthetic Dataset (A) and Ten-Task Synthetic Dataset (B).

## J  SUPPLEMENTARY RESULTS OF CBP ON HCP

We present the task classification accuracy for each subject on HCP, where CBP consistently outperforms chance level (Fig. 9). We provide supplementary confusion matrices of task classification for each subject (Fig. 10).

## K  SUPPLEMENTARY RESULTS OF CBP ON TLAE

We present the correlations between "student–student" alignment and learning performance for each connectivity pattern (Fig. 11A), as well as the correlations between "student–expert" alignment and learning performance for each connectivity pattern (Fig. 11B).

## L  LIMITATIONS AND FUTURE WORK

**Limitations and Future Work:** i) CBP models cognitive bases and their interactions via linear combinations, which may limit flexibility in capturing complex neural dynamics. Extending CBP with nonlinear formulations (e.g., Gaussian kernels) could enhance expressiveness but raises computational and interpretability challenges. ii) CBP currently focuses on fMRI-based whole-brain dynamics; integrating EEG with source localization would enable cross-modal modeling and a more comprehensive characterization of brain dynamics.

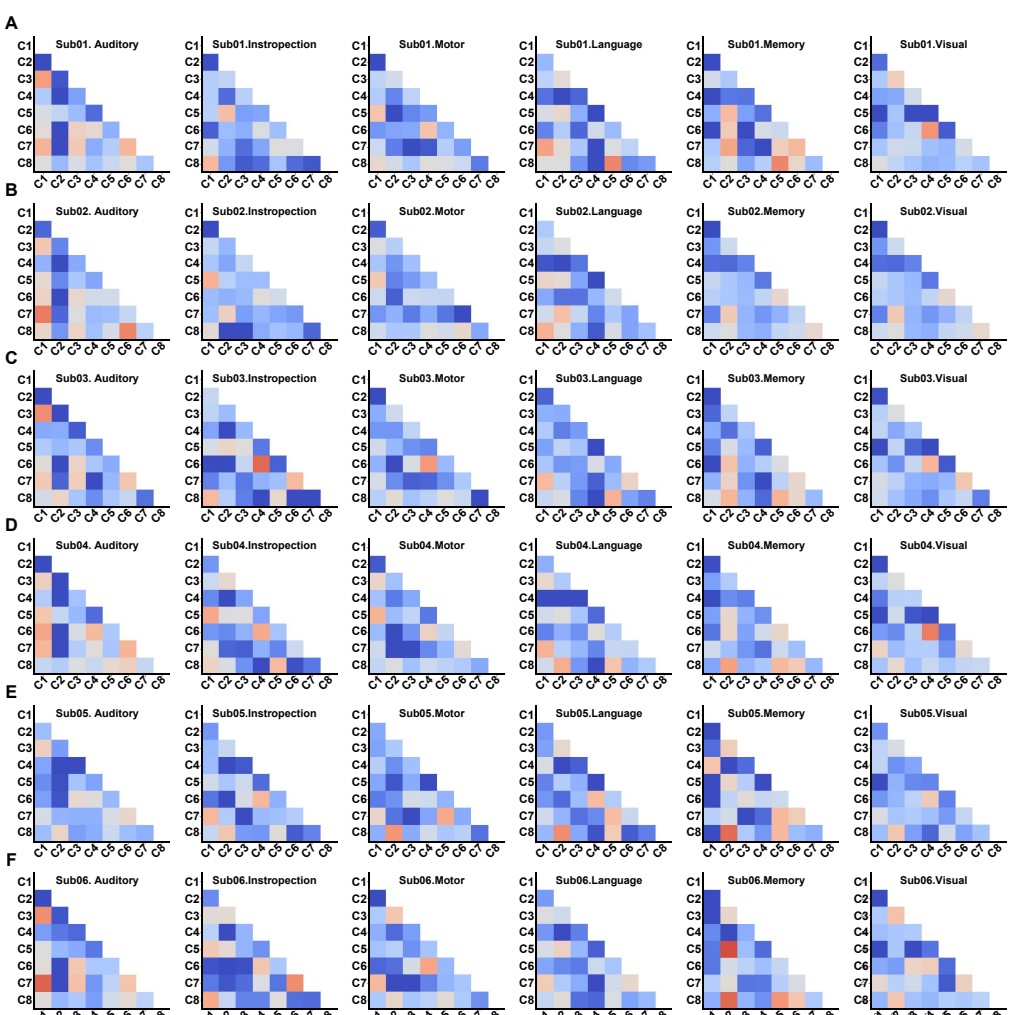

Figure 8: Coefficient ($C_p = \{c_{m,t}^{p,(s)}\}_{t=1}^{T_m^s}$) correlation matrix for each task and subject.

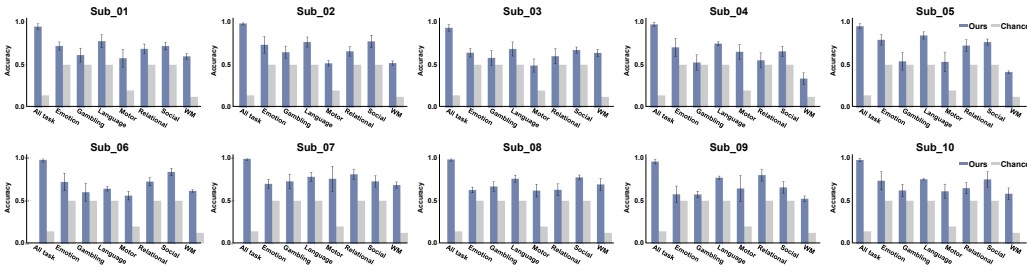

Figure 9: Task classification accuracy for each subject on HCP: CBP outperforms chance level.

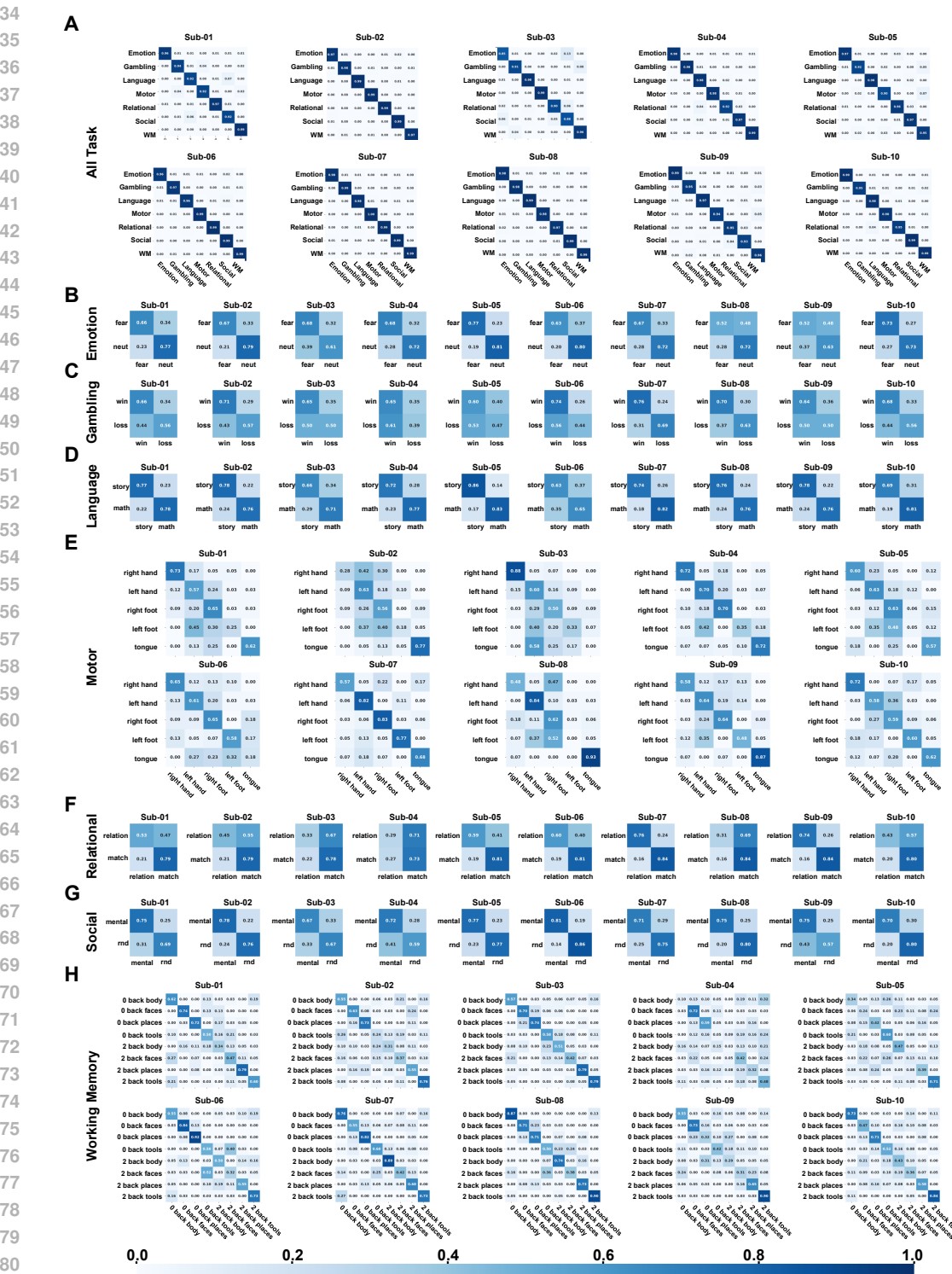

Figure 10: Confusion matrices across categories for each subject.

## M  SIMULATION SYSTEM WITH TUNABLE NONLINEARITY

To evaluate how well the proposed linear-by-parts formulation can capture nonlinear neural dynamics, we use a controlled synthetic system where the nonlinearity can be adjusted explicitly:

$$y = x^\tau, \tag{1}$$

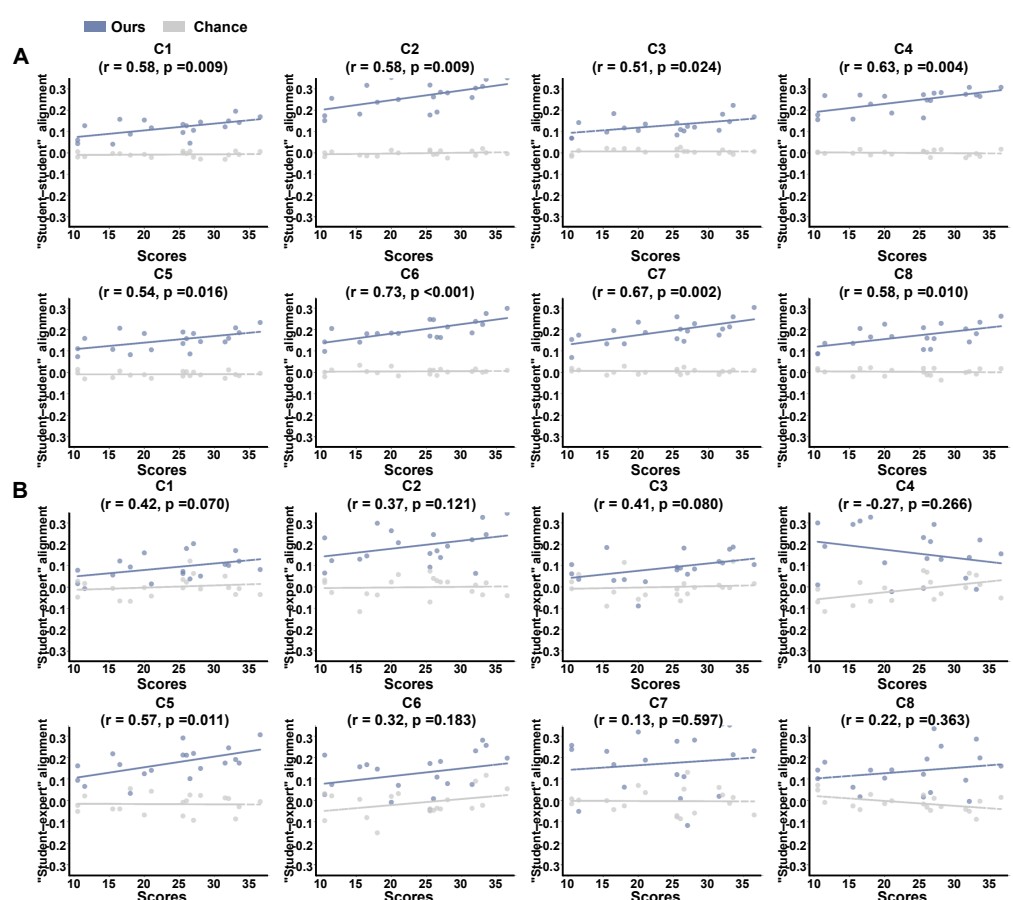

Figure 11: (A) Correlations between "student–student" alignment and learning performance for each connectivity pattern. (B) Correlations between "student–expert" alignment and learning performance for each connectivity pattern.

where the input $x$ is sampled as 400 evenly spaced points over the interval $[-2, 2]$; the exponent $\tau$ directly controls the degree of nonlinearity. This system is widely used in nonlinear system analysis because: (1) it preserves input variance and does not collapse to a fixed point; (2) it allows continuous tuning of nonlinearity strength; (3) it provides a clean testbed to evaluate piecewise-linear approximators.

# N   QUANTITATIVE EVALUATIONS OF INTERPRETABILITY

To ensure objectivity and reproducibility, we introduce three quantitative interpretability metrics that establish a principled mapping between the learned bases and the canonical Yeo-17/Yeo-7 functional networks. These metrics evaluate (i) spatial alignment, (ii) functional specificity, and (iii) global coverage of large-scale networks.

**1. Basis–Network Dice Overlap (alignment).** This metric quantifies how strongly a learned basis overlaps with each Yeo-17/Yeo-7 functional network, thus replacing qualitative matching with a rigorous numerical measure. For the $b$-th basis, $A^b = (A_{1,b}, \ldots, A_{N,b})$ denotes its weights across cortical ROIs, where $N = 360$. We first compute an activation threshold that selects its top $15\%$ highest-weighted ROIs. This threshold is given by the 85th percentile of its weight vector:

$$\theta_b = \text{Percentile}(A^{(b)}, 85). \tag{2}$$

We then identify the top-15% activation set of basis $b$ as:

$$S_b = \{i \mid A_{i,b} \geq \theta_b\}. \tag{3}$$

Let $R_k$ denote the ROI set of the $e$-th Yeo-17/Yeo-7 network. The Dice overlap is

$$D_{b,e} = \frac{2\,|S_b \cap R_e|}{|S_b| + |R_e|}, \tag{4}$$

where $e = 1, ..., E$, $E = 17$ or $E = 7$. A higher Dice value indicates stronger alignment between basis $b$ and network $e$. As shown in the Figs. 4F-G, by computing the Dice overlap between each basis and the Yeo-17/Yeo-7 functional networks, we obtain a basis–network mapping matrix, revealing strong correspondence between OT-derived bases and canonical networks.

**2. Basis Concentration (functional specificity).** This metric determines whether a basis is specialized (localized in one or a few networks) or integrative (distributed across multiple networks). We first normalize the Dice scores:

$$p_{b,e} = \frac{D_{b,e}}{\sum_{e'} D_{b,e'}}. \tag{5}$$

and then compute the Shannon entropy:

$$H_b = -\sum_e p_{b,e} \log p_{b,e}, \tag{6}$$

The concentration index is defined as:

$$C_b = 1 - \frac{H_b}{\log E}. \tag{7}$$

$$\text{Concentration} = \frac{1}{K} \sum_{b=1}^{K} C_b. \tag{8}$$

where $b = 1, ..., K$, $K = 12$. High Concentration reflects strong functional specificity. As shown in the Figs. 4H, we find that the bases learned from OT exhibit higher concentration compared to the bases trained directly on HCP.

**3. Functional Network Coverage (global completeness).** This metric assesses whether the full set of bases collectively covers all major functional systems, preventing collapse into a few high-variance networks. For each Yeo-17 network $e$, we compute the maximum normalized Dice:

$$m_e = \max_b p_{b,e}. \tag{9}$$

A network is considered sufficiently represented when this value exceeds a predefined threshold:

$$\delta_e = \begin{cases} 1, & m_e \geq \tau, \\ 0, & m_e < \tau, \end{cases} \tag{10}$$

The overall coverage is then defined as:

$$\text{Coverage} = \frac{1}{E} \sum_{e=1}^{E} \delta_e. \tag{11}$$

Higher coverage indicates that the learned bases collectively capture the full spectrum of large-scale functional networks, which reflects the completeness and robustness of the bases. As shown in the Figs. 4I, we find that the bases learned from OT exhibit higher coverage compared to the bases trained directly on HCP.

In summary, the learned bases quantitatively align with canonical functional systems, exhibit meaningful specificity, and achieve broad functional coverage. This grounding ensures that both the bases and the derived connectivity patterns are neuroscientifically interpretable in a principled and reproducible manner.

## O   COMPARISON OF CBP AND DEEP LEARNING BASELINES

Under the exact same data scale and task setting as CBP, we trained deep learning baselines from scratch, ensuring a fair comparison that does not rely on large-scale pretraining. The comparison results are shown in Table 8.

Table 8: Performance comparison (MSE and Pearson correlation) of CBP and baseline models on multiple datasets for reconstructing the observed signals $Y$ and identifying the cognitive bases $A$, latent trajectories $X$.

| Dataset | Three-Task Synthetic Dataset | | | Ten-Task Synthetic Dataset | | | OT Dataset | HCP Dataset | TLAE Dataset |
|---|---|---|---|---|---|---|---|---|---|
| Metric | $MSE(Y,\hat{Y})$ | $MSE(A,\hat{A})$ | $MSE(X,\hat{X})$ | $MSE(Y,\hat{Y})$ | $MSE(A,\hat{A})$ | $MSE(X,\hat{X})$ | $MSE(Y,\hat{Y})$ | $MSE(Y,\hat{Y})$ | $MSE(Y,\hat{Y})$ |
| ltrRNN | $0.1151 \pm 0.0013$ | / | / | $0.0809 \pm 0.0007$ | / | / | $0.0240 \pm 0.0012$ | $0.0131 \pm 0.0008$ | $0.0218 \pm 0.0014$ |
| AMAG | $0.0044 \pm 0.0002$ | / | / | $0.0032 \pm 0.0004$ | / | / | $0.0012 \pm 0.0002$ | $0.0063 \pm 0.0003$ | $0.0021 \pm 0.0002$ |
| NetFormer | $0.0520 \pm 0.0005$ | / | / | $0.0128 \pm 0.0011$ | / | / | $0.0086 \pm 0.0007$ | $0.0161 \pm 0.0008$ | $0.0140 \pm 0.0005$ |
| HGFM (2025) | $0.0094 \pm 0.0014$ | / | / | $0.0032 \pm 0.0008$ | / | / | $0.0061 \pm 0.0010$ | $0.0110 \pm 0.0005$ | $0.0095 \pm 0.0006$ |
| BrainMoE (2025) | $0.0051 \pm 0.0003$ | / | / | $0.0030 \pm 0.0002$ | / | / | $0.0045 \pm 0.0002$ | $0.0065 \pm 0.0018$ | $0.0088 \pm 0.0008$ |
| CBP | $\mathbf{0.0052} \pm 0.0002$ | $\mathbf{0.0041} \pm 0.0003$ | $\mathbf{0.0095} \pm 0.0002$ | $\mathbf{0.0021} \pm 0.0003$ | $\mathbf{0.0081} \pm 0.0002$ | $\mathbf{0.0092} \pm 0.0002$ | $\mathbf{0.0016} \pm 0.0003$ | $\mathbf{0.0075} \pm 0.0002$ | $\mathbf{0.0023} \pm 0.0002$ |

| Dataset | Three-Task Synthetic Dataset | | | Ten-Task Synthetic Dataset | | | OT Dataset | HCP Dataset | TLAE Dataset |
|---|---|---|---|---|---|---|---|---|---|
| Metric | $\rho(Y,\hat{Y})$ | $\rho(A,\hat{A})$ | $\rho(X,\hat{X})$ | $\rho(Y,\hat{Y})$ | $\rho(A,\hat{A})$ | $\rho(X,\hat{X})$ | $\rho(Y,\hat{Y})$ | $\rho(Y,\hat{Y})$ | $\rho(Y,\hat{Y})$ |
| ltrRNN | $0.3051 \pm 0.0001$ | / | / | $0.4090 \pm 0.0004$ | / | / | $0.4558 \pm 0.0007$ | $0.3025 \pm 0.0006$ | $0.3287 \pm 0.0004$ |
| AMAG | $0.9869 \pm 0.0003$ | / | / | $0.9272 \pm 0.0004$ | / | / | $0.9263 \pm 0.0003$ | $0.7984 \pm 0.0007$ | $0.9097 \pm 0.0002$ |
| NetFormer | $0.6784 \pm 0.0006$ | / | / | $0.6120 \pm 0.0015$ | / | / | $0.7738 \pm 0.0011$ | $0.3882 \pm 0.0022$ | $0.7799 \pm 0.0003$ |
| HGFM (2025) | $0.9227 \pm 0.0039$ | / | / | $0.9085 \pm 0.0116$ | / | / | $0.8634 \pm 0.0198$ | $0.7207 \pm 0.0181$ | $0.8751 \pm 0.0177$ |
| BrainMoE (2025) | $0.9794 \pm 0.0110$ | / | / | $0.9031 \pm 0.0014$ | / | / | $0.9059 \pm 0.0023$ | $0.7804 \pm 0.0369$ | $0.8992 \pm 0.0006$ |
| CBP | $\mathbf{0.9711} \pm 0.0002$ | $\mathbf{0.9403} \pm 0.0003$ | $\mathbf{0.9925} \pm 0.0002$ | $\mathbf{0.9478} \pm 0.0003$ | $\mathbf{0.8875} \pm 0.0004$ | $\mathbf{0.9678} \pm 0.0003$ | $\mathbf{0.9116} \pm 0.0002$ | $\mathbf{0.7617} \pm 0.0003$ | $\mathbf{0.9067} \pm 0.0002$ |

## P  STATISTICAL TESTS FOR CBP AND BRAINMOE ON MULTIPLE DATASETS

We conduct Wilcoxon signed-rank tests. Statistical results (Table 9) show that CBP outperforms BrainMoE on datasets where the difference is statistically significant, and performs comparably on datasets where the numerical differences are not significant. However, achieving competitive predictive accuracy is not our ultimate goal. Our central objective is to identify interpretable cognitive bases and connectivity patterns, thereby providing transparent and scientifically meaningful insights for neuroscience applications.

Table 9: Statistical tests for CBP and BrainMoE on multiple datasets.

| Metric | $MSE(Y,\hat{Y})$ | | |
|---|---|---|---|
| **Dataset** | **BrainMoE** | **CBP** | **p** |
| Three-Task Synthetic Dataset | $0.0051 \pm 0.0003$ | $0.0052 \pm 0.0002$ | 0.2324 |
| Ten-Task Synthetic Dataset | $0.0030 \pm 0.0002$ | $0.0021 \pm 0.0003$ | 0.0020 |
| OT dataset | $0.0045 \pm 0.0002$ | $0.0016 \pm 0.0003$ | 0.0020 |
| HCP dataset | $0.0065 \pm 0.0018$ | $0.0075 \pm 0.0002$ | 0.1602 |
| TLAE dataset | $0.0088 \pm 0.0008$ | $0.0023 \pm 0.0002$ | 0.0020 |
| Metric | $\rho(Y,\hat{Y})$ | | |
| **Dataset** | **BrainMoE** | **CBP** | **p** |
| Three-Task Synthetic Dataset | $0.9794 \pm 0.0110$ | $0.9711 \pm 0.0002$ | 0.0840 |
| Ten-Task Synthetic Dataset | $0.9031 \pm 0.0014$ | $0.9478 \pm 0.0003$ | 0.0023 |
| OT dataset | $0.9059 \pm 0.0023$ | $0.9116 \pm 0.0002$ | 0.0016 |
| HCP dataset | $0.7804 \pm 0.0369$ | $0.7617 \pm 0.0003$ | 0.1055 |
| TLAE dataset | $0.8992 \pm 0.0006$ | $0.9067 \pm 0.0002$ | 0.0019 |

## Q  THE DEFINITION OF MEAN(X) AND MEAN(C)

For Fig. 3B and Fig. 3D, the category-wise mean activations are computed by averaging across all subjects, all tasks belonging to the same cognitive category, and all time points. The OT dataset contains 103 task conditions, which are grouped into six canonical cognitive categories following prior work. For a given category, let $\mathcal{M}'$ denote the set of tasks in that category; $x_{m,t}^{i,(s)}$ the $i$-th latent state at time $t$ for subject $s$ in task $m$; and $c_{m,t}^{i,(s)}$ the activation coefficient of connectivity pattern $i$. The mean latent activation shown in Fig. 3B is computed as:

$$Mean(X_i) = \frac{\sum_{m \in \mathcal{M}'} \sum_{s=1}^{S} \sum_{t=1}^{T_m^{(s)}} x_{m,t}^{i,(s)}}{\sum_{m \in \mathcal{M}'} \sum_{s=1}^{S} T_m^{(s)}}. \tag{12}$$

and the mean connectivity-pattern activation in Fig. 3D is computed analogously using:

$$Mean(C_i) = \frac{\sum_{m \in \mathcal{M}'} \sum_{s=1}^{S} \sum_{t=1}^{T_m^{(s)}} c_{m,t}^{i,(s)}}{\sum_{m \in \mathcal{M}'} \sum_{s=1}^{S} T_m^{(s)}}. \tag{13}$$

This procedure ensures that each category-level curve reflects the aggregated activation across all tasks, subjects, and time points belonging to that category.

## R    QUALITATIVE INTERPRETATION OF REGULARIZATION TERMS

The regularization terms in CBP are not heuristic add-ons, but are carefully designed to reflect the structural properties of fMRI data. Each term has a clear motivation and plays a distinct role in ensuring stability and interpretability. Specifically:

(1) Laplacian regularization: promoting spatial coherence and cross-task generalization. This term encourages brain regions with similar temporal evolution patterns to be grouped into the same basis, forming coherent spatial clusters. It offers two key benefits: (i) **Suppressing local high-dimensional noise**, preventing each basis from becoming spatially fragmented; (ii) **Enhancing cross-task generalization**, encouraging the model to capture stable shared structures rather than overfitting to task-specific fluctuations. Ablation experiments show that removing this regularization term leads to noticeably noisier and less coherent bases.

(2) Sparsity regularization: promoting functional specificity. Sparsity encourages each basis to focus on a limited subset of brain regions, forming meaningful and functionally specific subnetworks. Without sparsity, each basis tends to diffuse across the whole brain, losing functional specialization and reducing interpretability.

(3) Temporal smoothness regularization: enforcing physiologically realistic dynamics. BOLD signals evolve slowly over time due to the hemodynamic response, and do not exhibit rapid, high-frequency oscillations. Temporal smoothness removes abrupt fluctuations and enforces temporally coherent, physiologically plausible interaction patterns. Ablation shows that without this term, the temporal patterns become unstable or noisy, deviating from realistic fMRI dynamics.

(4) Decorrelation regularization: preventing redundancy between patterns. This term ensures that each pattern captures complementary structure instead of duplicating others. Removing it leads to highly redundant patterns and reduced interpretability.

Taken together, the four regularizers constrain **spatial coherence, cross-task generalization, functional specificity, temporal stability, and non-redundancy among patterns**, each addressing a distinct property of fMRI data. Ablations confirm that their contributions are independent and non-interchangeable. Overall, they ensure the stability, generalizability, and spatial–temporal interpretability of CBP, demonstrating that the model design is principled rather than ad hoc.

