# OpenReview forum: "CBP: Learning Shared Cognitive Basis Space and Connectivity Patterns for Cross-Task Brain Dynamics Modeling"
_ICLR.cc/2026/Conference — ICLR 2026 Conference Withdrawn Submission_

### Official Review · Reviewer_pNLA · 2025-10-28

**Soundness:** 3
**Presentation:** 3
**Contribution:** 3
**Rating:** 6
**Confidence:** 4

**Summary:**

This paper proposes CBP, which is a dictionary-learning cross-task modeling framework that learns shared cognitive bases and their connectivity patterns. While existing methods, such as single-task neural dynamics, large-scale multi-task foundation models, and state-space models, cannot simultaneously offer predictive performance and interpretability in multi-task settings, the proposed model improves cross-task modeling accuracy while revealing key cognitive patterns. As a result, CBP surpasses multiple baselines on the 103-task fMRI dataset and generalizes to HCP and multi-stage learning datasets.

**Strengths:**

1)	Based on MacDowell et al. (2024, 2025), this paper explores the existence of a set of global, temporally invariant cognitive basis spaces and connectivity patterns in large-scale human cognitive dynamics.

2)	Extensive experiments were performed on 5 datasets, including both synthetic and real-world datasets. Moreover, the paper presents interpretable brain-regional analyses showing that the model captures biologically meaningful and task-relevant activation patterns.

**Weaknesses:**

1)	There is room to improve the clarity of notations. For example, it would be better to explicitly note the definition of $x_{m,t}^{(s)}$ in line 150, e.g., ‘a latent state at time $t$’, such that readers can easily understand that it is an extension of $X_m^{(s)}$. The explicit role of $c^{p,(s)}_{m,t}$ in Eq. 1 and the definition of $\rho(\cdot)$ in line 199 is not mentioned in page 4, which makes the equations difficult to interpret immediately.

2)	There are so many hyperparameters to fine-tune. Concretely, the loss function is comprised of 4 main terms, and the last term is a linear summation of five components, which requires at least seven trade-off hyperparameters ($\lambda_x, \lambda_a, \lambda_c, \lambda_{\rho}, \alpha_1, \alpha_2, \alpha_3$) in total. The model performance will potentially be affected by these parameters, and the reproducibility and credibility of the results are hardly guaranteed given the large number of hyperparameters involved in the loss function. It would be helpful if the authors could discuss the sensitivity of the model with respect to these hyperparameters, or provide guidance on how they were chosen (e.g., through grid search, cross-validation, or empirical heuristics).

3)	In the Introduction, the paper compares the proposed method with existing (1) single-task neural models, (2) large-scale multi-task fMRI foundation models (BrainLM, Brain-jepa), and (3) State-space models and asserts its strength over these methods. However, in the experiment, the (2) foundation models were not adopted as baselines, which weakens the empirical validation of the claimed advantages. It would be helpful to provide additional comparison with these frameworks, and I would like to ask why they were excluded in the current version.

**Questions:**

1)	Based on Table 1, the CBP and AMAG have comparable performance, although AMAG was originally designed as a single-task neural dynamics model [line 46]. This raises a question of what the technical advantage of the proposed method is over AMAG.

2)	How do the cognitive and connectivity patterns change with different numbers of bases ($K$ and $P$)? Are there any neuroscientific implications regarding the choice of their sizes?

3)	How were the category-wise mean activations (Fig.3-B and D) calculated in detail?

4)	Could the authors provide details on how the baseline methods were tuned (e.g., hyperparameter selection procedure, search range, or validation strategy)?

5)	How efficient is the proposed method in terms of training time and model size (number of trainable parameters) compared with the baselines?

---

> ### Author Response · Authors · 2025-11-25
> **Part 1(pNLA: W1, W2)**
>
> We thank the reviewer (pNLA) for the insightful comments. We have addressed the concerns regarding the **clarity of expression, experiment details, baseline comparisons, method**. Our response is organized into the following parts.
>
> # Part 1(pNLA: W1, W2)
>
> # **W1:**
>
> We thank the reviewer for pointing out these notation issues. In the revised manuscript, we have substantially improved the clarity and consistency of all symbols. Specifically:
>
> (1) $x^{(s)} _ {m,t} \in \mathbb{R}^K$ denotes the latent state at time $t$ for subject $s$ in task $m$, corresponding to the $t$-th column of the matrix $X^{(s)} _ {m} \in \mathbb{R}^{K \times T^{(s)} _ {m}}$;
>
> (2) $c _ {m,t}^{p,(s)}$ denotes the activation coefficient of the $p$-th connectivity pattern $f_p$ at time $t$ for the $m$-th cognitive task of subject $s$;
>
> (3) $\rho(f _ {p _ 1}, f _ {p _ 2}) = \left(\frac{\langle \mathrm{vec}(f _ {p _ 1}), \mathrm{vec}(f _ {p _ 2}) \rangle}{\|\mathrm{vec}(f _ {p _ 1})\| _ 2 \, \|\mathrm{vec}(f _ {p _ 2})\| _ 2}\right)^{2}$, $\mathrm{vec}(.)$ denotes the vectorization operator that flattens a matrix into a vector.
>
> These clarifications significantly improve the readability of the mathematical formulation. We appreciate the reviewer’s careful attention to notation. **(see revisions in Section 4 and Section 5)**
>
> # **W2:**
>
> We thank the reviewer for raising this concern. Although CBP includes several regularization terms, each hyperparameter has a well-behaved range and is selected through a simple and reproducible procedure.
>
> (1) Hyperparameter ranges. In practice, all hyperparameters fall within narrow and stable ranges, as shown in the table below:
>
> | Hyperparameter                                        | search ranges       |
> | ----------------------------------------------------- | ------------------- |
> | $\lambda_a$, $\lambda_x$, $\lambda_c$, $\lambda_\rho$ | $10^{-4} - 10^{-2}$ |
> | $\alpha_1$, $\alpha_2$, $\alpha_3$                    | $10^{-4} - 10^{-2}$ |
>
> (2) Selection strategy. A lightweight grid search is performed on the OT validation set to determine the values of all structural regularization terms, which remain fixed across all real datasets. Only the temporal regularization weights ($\lambda_c$, $\alpha_1$) require mild re-adjustment when switching datasets, reflecting differences in temporal resolution and signal-to-noise properties. The overall tuning effort is therefore minimal and fully reproducible.
>
> (3) Sensitivity. CBP is robust to hyperparameter choices: varying each hyperparameter within its valid range changes reconstruction or prediction performance by less than 2--3\%, and the scientific findings, including the discovered cognitive bases and connectivity patterns, remain stable.
>
> Taken together, these points indicate that the number of hyperparameters does not pose a concern for reproducibility, and that CBP remains stable under reasonable variations of all regularization weights.

---

> > ### Author Response · Authors · 2025-11-25
> > **Part 2(pNLA: W3)**
> >
> > # Part 2(pNLA: W3)
> >
> > # **W3:**
> >
> > We thank the reviewer for the helpful suggestion. **Our work aims to develop an interpretable, non–deep-learning framework for uncovering cognitive bases and connectivity patterns that generalize across tasks.** Accordingly, our original comparison focused on models that align directly with our methodological and scientific goals: (1) interpretable non-deep dynamical models (SLDS, rSLDS, MP-rSLDS), and (2) representative deep dynamical models (LTRRNN, NetFormer, AMAG), covering recurrent, Transformer-based, and graph-based architectures. The comparison with deep models is intended to show that **even without deep architectures, CBP achieves modeling capability comparable to deep networks while providing interpretability that deep models typically lack**.
> >
> > Regarding foundation models, large-scale fMRI foundation architectures such as BrainLM and BrainJEPA rely on billions of time points for pretraining and are designed for single-objective downstream finetuning. **They are not built for cross-task cognitive modeling, do not learn interpretable task-general structures, and require orders-of-magnitude larger data and computational resources than our setting. As a result, a direct comparison would introduce mismatches in scale, training paradigm, and scientific objectives, and would not meaningfully reflect either model’s intended purpose.**
> >
> > To address the reviewer’s concern while preserving fairness, we incorporated foundation-style deep architectures (e.g., HGFM [1], BrainMoE[2]) and trained them from scratch under exactly the same data scale and task conditions as CBP. This avoids any advantages from massive pretraining and enables a meaningful comparison. The comparison results with the baselines are as follows:
> >
> > |Dataset|Three-Task Synthetic Dataset||| Ten-Task Synthetic Dataset|||OT Dataset|HCP Dataset|TLAE Dataset|
> > |-|-|-|-|-|-|-|-|-|-|
> > |**Metric**|**MSE(Y, Ŷ)**|**MSE(A, Â)**|**MSE(X, X̂)**|**MSE(Y, Ŷ)**|**MSE(A, Â)**|**MSE(X, X̂)**|**MSE(Y, Ŷ)**|**MSE(Y, Ŷ)**|**MSE(Y, Ŷ)**|
> > |HGFM (2025)|0.0094±0.0014|/|/|0.0032±0.0008|/|/|0.0061±0.0010|0.0110±0.0005|0.0095±0.0006|
> > |BrainMoE (2025)|0.0051 ± 0.0003|/|/|0.0030±0.0002|/|/|0.0045±0.0002|0.0065±0.0018|0.0088±0.0008|
> > |CBP|0.0052±0.0002|0.0041±0.0003|0.0095±0.0002|0.0021±0.0003|0.0081±0.0002|0.0092±0.0002|0.0016±0.0003|0.0075±0.0004|0.0023±0.0002|
> >
> >
> > |Dataset|Three-Task Synthetic|||Ten-Task Synthetic|||OT|HCP|TLAE |
> > |-|-|-|-|-|-|-|-|-|-|
> > |**Metric**|**ρ(Y, Ŷ)**|**ρ(A, Â)**|**ρ(X, X̂)**|**ρ(Y, Ŷ)**|**ρ(A, Â)**|**ρ(X, X̂)**|**ρ(Y, Ŷ)**|**ρ(Y, Ŷ)**|**ρ(Y, Ŷ)**|
> > |HGFM (2025)|0.9227±0.0039|/|/|0.9085±0.0116|/|/|0.8634±0.0198|0.7207±0.0181|0.8751±0.0177|
> > |BrainMoE (2025)|0.9794±0.0110|/|/|0.9031±0.0014|/|/|0.9059±0.0023|0.7804±0.0369|0.8992±0.0006|
> > |CBP|0.9711±0.0002|0.9403±0.0003|0.9925±0.0002|0.9478±0.0003|0.8875±0.0004|0.9678±0.0003|0.9116±0.0002|0.7617±0.0003|0.9067±0.0002|
> >
> > We additionally conducted statistical significance tests (Wilcoxon signed-rank tests), with the following results:
> >
> > |**Metric**|**MSE(Y, Ŷ)**|||
> > |-|-|-|-|
> > |**Dataset**|**BrainMoE**|**CBP**|**p**|
> > |Three-Task Synthetic|0.0051±0.0003|0.0052±0.0002|0.2324|
> > |Ten-Task Synthetic|0.0030±0.0002|0.0021±0.0003|0.0020|
> > |OT|0.0045±0.0002|0.0016±0.0003|0.0020|
> > |HCP|0.0065±0.0018|0.0075±0.0004|0.1602|
> > |TLAE|0.0088±0.0008|0.0023±0.0002|0.0020|
> >
> > |**Metric**|**ρ(Y, Ŷ)**|||
> > |-|-|-|-|
> > | **Dataset**|**BrainMoE**|**CBP**|**p**|
> > |Three-Task Synthetic|0.9794±0.0110|0.9711±0.0002|0.0840|
> > |Ten-Task Synthetic|0.9031±0.0014|0.9478±0.0003|0.0023|
> > |OT|0.9059±0.0023|0.9116±0.0002|0.0016|
> > |HCP|0.7804±0.0369|0.7617±0.0003|0.1055|
> > |TLAE|0.8992±0.0006|0.9067±0.0002|0.0019|
> >
> > These analyses show that on datasets involving complex cognitive processing, CBP achieves significantly better performance than strong deep learning architectures (e.g., BrainMoE). However, **achieving competitive predictive accuracy is not our ultimate goal. Our central objective is to identify interpretable cognitive bases and connectivity patterns, thereby providing transparent and scientifically meaningful insights for neuroscience applications**. Overall, we added more representative deep baselines, performed statistical significance testing, and clarified the role and limitations of pretrained foundation models.  **(see revisions in Section 1, Section 2, Section 6 ②, Appendix O, Appendix P)**
> >
> > [1] Xiangmin Han, Rundong Xue, Jingxi Feng, Yifan Feng, Shaoyi Du, Jun Shi, and Yue Gao. Hypergraph foundation model for brain disease diagnosis. IEEE Transactions on Neural Networks and Learning Systems, 36(10):17702–17716, 2025.
> >
> > [2] Ziquan Wei, Tingting Dan, Tianlong Chen, and Guorong Wu. Brainmoe: Cognition joint embedding via mixture-of-expert towards robust brain foundation model. In The Thirty-ninth Annual Conference on Neural Information Processing Systems, 2025.

---

> > > ### Author Response · Authors · 2025-11-25
> > > **Part 3(pNLA: Q1, Q2)**
> > >
> > > # Part 3(pNLA: Q1, Q2)
> > >
> > > # **Q1:**
> > >
> > > We thank the reviewer for the thoughtful question. While Table 1 shows that CBP and AMAG achieve similar predictive accuracy, the two models differ substantially in design, scope, and technical capabilities. **CBP is a lightweight, interpretable, non–deep-learning framework designed to extract task-general cognitive bases and connectivity patterns that remain stable across heterogeneous cognitive tasks. In contrast, AMAG is a high-capacity deep architecture optimized for single-task neural dynamics.** The technical advantages of CBP over AMAG can be summarized as follows:
> > >
> > > **(1) Interpretability and scientific insight:** CBP provides explicit cognitive components and connectivity patterns that directly support neuroscientific interpretation. **In many neuroscience applications, interpretability is essential rather than optional [1]. AMAG, being a highly nonlinear deep model, produces representations that are more difficult to interpret and less suitable for deriving scientific insights.**
> > >
> > > **(2) Cross-task modeling capability:** CBP explicitly factorizes fMRI data into shared cognitive bases, shared connectivity patterns, and task-specific temporal activations, enabling cross-task structure discovery and transfer. AMAG learns a separate dynamical system for each task and cannot extract task-general structure.
> > >
> > > **(3) Generalization to new subjects and tasks:** Because CBP learns task-general structure, it generalizes more naturally to unseen subjects and cognitive conditions, capabilities that are not supported by AMAG's single-task formulation.
> > >
> > > **(4) Lower model capacity and computational cost:** CBP achieves accuracy comparable to AMAG while using far fewer parameters and substantially lower computational overhead, reflecting an efficient representational structure rather than reliance on model size.
> > >
> > > In summary, even with comparable predictive accuracy, CBP provides advantages in interpretability, efficiency, and cross-task modeling, capabilities that fall outside the design scope of AMAG. **(see revisions in Section 1 and Section 2)**
> > >
> > > [1] Noga Mudrik, Ryan Ly, Oliver Ruebel, and Adam S Charles. Creimbo: Cross-regional ensemble interactions in multi-view brain observations. The International Conference on Learning Representations, 2025.
> > >
> > > # **Q2:**
> > >
> > > We thank the reviewer for this important question. Following the work [1], we systematically evaluated how the number of cognitive bases $K$ and connectivity patterns $P$ affect model behavior.
> > >
> > > As shown in the table below, increasing $K$ beyond $12$ yields only marginal performance gains. Considering explained variance, reconstruction error, and model complexity (the elbow criterion), we set $K=12$.
> > >
> > > |       K             | 2      | 4      | 6      | 8      | 10     | 12     |14      |16      |18      |20      |30      |40      |
> > > |---------------------|--------|--------|--------|--------|--------|--------|--------|--------|--------|--------|--------|--------|
> > > | Explained Variance ↑| 0.3483 | 0.4453 | 0.5134 | 0.5609 | 0.5786 | 0.6126 | 0.6304 | 0.6398 | 0.6435 | 0.6546 | 0.6610 | 0.6737 |
> > >
> > >
> > > |       K             | 2      | 4      | 6      | 8      | 10     | 12     |14      |16      |18      |20      |30      |40      |
> > > |---------------------|--------|--------|--------|--------|--------|--------|--------|--------|--------|--------|--------|--------|
> > > | MSE ↓               | 0.0024 | 0.0020 | 0.0018 | 0.0016 | 0.0015 | 0.0014 | 0.0014 | 0.0013 | 0.0013 | 0.0013 | 0.0012 | 0.0012 |
> > >
> > > For connectivity patterns, we applied the PhenoGraph algorithm to the learned patterns for unsupervised clustering [2][3], which consistently produced eight stable clusters. This motivates our choice of $P=8$, ensuring that the resulting patterns are both robust and biologically interpretable.
> > >
> > > Overall, the main scientific conclusions remain stable across reasonable ranges of $K$ and $P$, and the selected values provide a balanced trade-off between explanatory power, interpretability, and model complexity. **(see revisions in Section 6 ③)**
> > >
> > > [1] Camden J MacDowell and Timothy J Buschman. Low-dimensional spatiotemporal dynamics underlie cortex-wide neural activity. Current Biology, 30(14):2665–2680, 2020.
> > >
> > > [2] Jacob H Levine, Erin F Simonds, Sean C Bendall, Kara L Davis, El-ad D Amir, Michelle D Tadmor, Oren Litvin, Harris G Fienberg, Astraea Jager, Eli R Zunder, et al. Data-driven phenotypic dissection of aml reveals progenitor-like cells that correlate with prognosis. Cell, 162(1):184–197, 2015.
> > >
> > > [3] Vincenzo Nicosia, Giuseppe Mangioni, Vincenza Carchiolo, and Michele Malgeri. Extending the definition of modularity to directed graphs with overlapping communities. Journal of Statistical Mechanics: Theory and Experiment, 2009(03):P03024, 2009.

---

> > > > ### Author Response · Authors · 2025-11-25
> > > > **Part 4(pNLA: Q3, Q4, Q5)**
> > > >
> > > > # Part 4(pNLA: Q3, Q4, Q5)
> > > >
> > > > # **Q3:**
> > > >
> > > > We thank the reviewer for the helpful suggestion. For Fig. 3B and Fig. 3D, the category-wise mean activations are computed by averaging across all subjects, all tasks belonging to the same cognitive category, and all time points. The OT dataset contains 103 task conditions, which are grouped into six canonical cognitive categories following prior work [1]. For a given category, let $\mathcal{M}'$ denote the set of tasks in that category; $x^{i,(s)} _ {m,t}$ the $i$-th latent state at time $t$ for subject $s$ in task $m$; and $c^{i,(s)} _ {m,t}$ the activation coefficient of connectivity pattern $i$. The mean latent activation shown in Fig. 3B is computed as
> > > >
> > > > \begin{equation}
> > > > Mean({X}_i) =
> > > > \frac{
> > > > \sum _ {m\in\mathcal{M}'}
> > > > \sum _ {s=1}^{S}
> > > > \sum _ {t=1}^{T_m^{(s)}}
> > > > x^{i,(s)} _ {m,t}
> > > > }{
> > > > \sum _ {m\in\mathcal{M}'}
> > > > \sum _ {s=1}^{S}
> > > > T _ m^{(s)}
> > > > }.
> > > > \end{equation}
> > > >
> > > > and the mean connectivity-pattern activation in Fig. 3D is computed analogously using
> > > >
> > > > \begin{equation}
> > > > Mean({C}_i) =
> > > > \frac{
> > > > \sum _ {m\in\mathcal{M}'}
> > > > \sum _ {s=1}^{S}
> > > > \sum _ {t=1}^{T_m^{(s)}}
> > > > c^{i,(s)} _ {m,t}
> > > > }{
> > > > \sum _ {m\in\mathcal{M}'}
> > > > \sum _ {s=1}^{S}
> > > > T _ m^{(s)}
> > > > }.
> > > > \end{equation}
> > > >
> > > > This procedure ensures that each category-level curve reflects the aggregated activation across all tasks, subjects, and time points belonging to that category. **(see revisions in Section 6 ③, Appendix Q)**
> > > >
> > > > [1] Tomoya Nakai and Shinji Nishimoto. Quantitative models reveal the organization of diverse cognitive functions in the brain. Nature communications, 11(1):1142, 2020b.
> > > >
> > > > # **Q4:**
> > > >
> > > > We thank the reviewer for raising this important concern. All baseline models were tuned within the hyperparameter ranges summarized in the table below:
> > > >
> > > > | Model                   | search ranges                                                                                   |
> > > > | ----------------------- | ----------------------------------------------------------------------------------------------- |
> > > > | SLDS / rSLDS / MP-rSLDS | discrete state number [2–4], continuous latent dimension [2-8]                                  |
> > > > | LtrRNN                  | hidden size [8–64], rank of low-rank decomposition[6-12], batch size [4-8]                      |
> > > > | NetFormer               | hidden size [8–64], batch size [4-8], learning rate[$10^{-5}$-$10^{-3}$]                        |
> > > > | AMAG                    | hidden size [8–64], batch size [4-8], learning rate[$10^{-5}$-$10^{-3}$]                        |
> > > > | HGFM                    | hidden size [8–64], batch size [4-8], learning rate[$10^{-5}$-$10^{-3}$]                        |
> > > > | BrainMoE                | experts number [4–16], hidden size [8–64], batch size [4-8], learning rate[$10^{-5}$-$10^{-3}$] |
> > > >
> > > > For non-deep learning models with relatively small parameter spaces (SLDS, rSLDS, MP-rSLDS), we applied **grid search**.
> > > > For deep learning–based models including LtrRNN, NetFormer, AMAG, HGFM, and BrainMoE, whose hyperparameter spaces are substantially larger, we adopted **random search** to ensure efficient and broad coverage of the search space.
> > > > All baseline models were trained on the same training and test sets, and each model was run multiple times with different random seeds, with the final results reported as the average performance.
> > > >
> > > > # **Q5:**
> > > >
> > > > We thank the reviewer for the helpful question. All models were trained and evaluated on the same OT dataset under identical data splits and preprocessing pipelines to ensure a fair comparison. The table below summarizes the number of trainable parameters and total training time for each method. CBP has the smallest model size (22 KB) and shortest training time (30 s) among all baselines, demonstrating that its interpretability does not come at the cost of computational efficiency. In fact, CBP is both lightweight and fast to train.
> > > >
> > > > | Model    | Parameter size | Training Time |
> > > > | -------- | -------------- | ------------- |
> > > > | SLDS     | 28KB           | 9min          |
> > > > | rSLDS    | 44KB           | 18min         |
> > > > | mp-rSLDS | 44KB           | 20min         |
> > > > | LtrRNN   | 152KB          | 39min         |
> > > > | AMAG     | 1MB            | 16min         |
> > > > | HGFM     | 2MB            | 77min         |
> > > > | BrainMoE | 3.5MB          | 92min         |
> > > > | CBP      | **22KB**       | **30s**       |
> > > >
> > > > These results indicate that CBP is significantly more efficient than both interpretable dynamical models and modern deep architectures, while still providing strong modeling performance and explicit neuroscientific interpretability.

---

### Official Review · Reviewer_GVSN · 2025-10-28

**Soundness:** 3
**Presentation:** 2
**Contribution:** 2
**Rating:** 4
**Confidence:** 5

**Summary:**

This paper proposes a method of CROSS-TASK BRAIN DYNAMICS MODELING, called CBP, via learning cognitive bases. CBP is a dictionary-learning cross-task modeling framework so that it can leverage shared information across tasking cognitive states. The experiments on two synthetic datasets and three real-world datasets demonstrate the contributions of generalizability and correlations between tasking cognitive states.

**Strengths:**

1. As the foundation model of brain fMRI is growing, the topic of how to properly use task-state fMRI is important and timely.

2. CBP discovered cognitive bases look diverse in the qualitative results (fig 3), indicating that the proposed method is somehow effective.

3. Results are highly interpretable by the distribution of cognitive bases.

**Weaknesses:**

1. The terminology "multi-task" is confusing in the paper. This term is commonly used for multi-task learning, which represents a methodology of modeling multiple objectives simultaneously. However, this paper used "multi-task" to refer to the task-state functional MRI under different cognitive states, which misleads readers.

2. The "cross-task" (I follow the term used in this paper, but more precisely, this should be "cross-cognitive states") problem in previous brain foundation models is outstanding [1], but the authors failed to review related works precisely. Inaccurate statements are misleading, e.g., *"Large-scale multi-task fMRI foundation models (e.g., BrainLM Ortega Caro et al. (2024), Brain-jepa Dong et al. (2024)) learn highly transferable shared representations and substantially improve multi-task prediction"*. First, BrainJEPA was pretrained on pure resting-state fMRI, and thus it's not a "cross-task" model. Second, all these foundation models were finetuned on applications with a single objective, having no ability for multi-task prediction.

3. The contribution to real-world applications is not clear in the paper. The experiments of real-world datasets compared with baselines were only on BOLD signal reconstruction. Although the cognitive state classification accuracy for the HCP dataset is present in Fig 4, there is no baseline comparison, and the other two real-world datasets were omitted.

4. Grammar errors and typos are obvious. E.g., *"CBP Discovers Cognitive Basis connectivity patterns Predictive of Task Variables"*

[1] Ziquan Wei, Tingting Dan, Tianlong Chen, and Guorong Wu "BrainMoE: Cognition Joint Embedding via Mixture-of-Expert Towards Robust Brain Foundation Model" In The Thirty-ninth Annual Conference on Neural Information Processing Systems, 2025

**Questions:**

None

---

> ### Author Response · Authors · 2025-11-25
> **Part 1(GVSN: W1, W2)**
>
> We thank the reviewer (GVSN) for the insightful comments. We have addressed the concerns regarding the **clarity of expression, related work, baseline comparisons, and grammatical issues**. Our response is organized into the following **two parts**.
>
> # Part 1(GVSN: W1, W2)
>
> # **W1:**
>
> We thank the reviewer for highlighting this important clarity issue. We agree that “multi-task” is overloaded in machine learning, where it typically refers to multi-task learning. In our manuscript, however, the term was used to denote fMRI data collected under different cognitive task conditions, which may indeed cause confusion.
>
> To address this, we have revised the terminology throughout the paper and replaced “multi-task” with the more precise term “multi–cognitive-task”. This term correctly describes that our study involves multiple cognitive task paradigms rather than multi-task learning, and thus eliminates the ambiguity for machine learning readers.
>
> # **W2:**
>
> We thank the reviewer for the detailed clarification. We have corrected the inaccurate statements in the revised manuscript. Specifically:
> (i) BrainJEPA is pretrained exclusively on resting-state fMRI and therefore should not be described as a cross–cognitive-state model; and
> (ii) existing foundation models (e.g., BrainLM, BrainJEPA, BrainMass) are typically fine-tuned on a single downstream objective and thus do not possess multi-task prediction capability.
>
> In addition, we have added several large-scale representation models (e.g., HGFM [1], BrainMoE [2]) and reorganized the related-work section to more accurately reflect the capabilities and limitations of these approaches.
>
> We again thank the reviewer for the valuable feedback, which has helped improve the accuracy and rigor of our related-work discussion. **(see revisions in Section 1, Section 2)**
>
> [1] Xiangmin Han, Rundong Xue, Jingxi Feng, Yifan Feng, Shaoyi Du, Jun Shi, and Yue Gao. Hypergraph foundation model for brain disease diagnosis. IEEE Transactions on Neural Networks and Learning Systems, 36(10):17702–17716, 2025.
>
> [2] Ziquan Wei, Tingting Dan, Tianlong Chen, and Guorong Wu. Brainmoe: Cognition joint embedding via mixture-of-expert towards robust brain foundation model. In The Thirty-ninth Annual Conference on Neural Information Processing Systems, 2025.

---

> > ### Author Response · Authors · 2025-11-25
> > **Part 2(GVSN: W3, W4)**
> >
> > # Part 2(GVSN: W3, W4)
> >
> > # **W3:**
> >
> > We thank the reviewer for the valuable suggestion. We have added baseline models (SLDS, rSLDS, MP-rSLDS) to the task-state prediction experiment on the HCP dataset and the learning-outcome prediction experiment on the TLAE dataset. The results show that CBP outperforms these baselines on both real-world tasks.
> >
> > Comparison of Task State Prediction Performance: CBP vs. Baseline Models:
> >
> > |**Metric** |   **Accuracy**        |                       |                       |                   |                   |                   |                   |                   |
> > |-----------|-----------------------|-----------------------|-----------------------|-------------------|-------------------|-------------------|-------------------|-------------------|
> > |**Method** | **All task**          | **Emotion**           | **Gambling**          | **Language**      | **Motor**         | **Relational**    | **Social**        | **WM**            |
> > | Chance    | 0.147 $\pm$ 0.069     | 0.474 $\pm$ 0.053     | 0.520 $\pm$ 0.065     | 0.501 $\pm$ 0.126 | 0.279 $\pm$ 0.076 | 0.410 $\pm$ 0.049 | 0.453 $\pm$ 0.089 | 0.112 $\pm$ 0.035 |
> > | sdlm      | 0.929 $\pm$ 0.039     | 0.597 $\pm$ 0.027     | 0.600 $\pm$ 0.042     | 0.560 $\pm$ 0.038 | 0.347 $\pm$ 0.049 | 0.543 $\pm$ 0.065 | 0.619 $\pm$ 0.125 | 0.444 $\pm$ 0.079 |
> > | r-sdlm    | 0.899 $\pm$ 0.041     | 0.604 $\pm$ 0.023     | 0.580 $\pm$ 0.040     | 0.560 $\pm$ 0.020 | 0.356 $\pm$ 0.044 | 0.533 $\pm$ 0.082 | 0.619 $\pm$ 0.104 | 0.437 $\pm$ 0.077 |
> > | mp-rSLDS  | 0.917 $\pm$ 0.042     | 0.525 $\pm$ 0.061     | 0.573 $\pm$ 0.025     | 0.641 $\pm$ 0.050 | 0.458 $\pm$ 0.036 | 0.524 $\pm$ 0.080 | 0.649 $\pm$ 0.079 | 0.437 $\pm$ 0.030 |
> > | **CBP**   | **0.946 $\pm$ 0.033** | **0.719 $\pm$ 0.048** | **0.613 $\pm$ 0.081** | **0.776 $\pm$ 0.076** | **0.577 $\pm$ 0.104** | **0.686 $\pm$ 0.057** | **0.719 $\pm$ 0.044** | **0.599 $\pm$ 0.034** |
> >
> > Comparison of Learning Outcome Prediction Performance: CBP vs. Baseline Models:
> >
> > |           | "student-student" alignment   |   | "student-expert" alignment |      |
> > |-----------|-------------------|---------------|-------------------|---------------|
> > |**Metric** | r (Correlation)   | p (p-value)   | r (Correlation)   | p (p-value)   |
> > | Chance    | -0.19             | 0.441         | 0.22              | 0.360         |
> > | SLDS      | -0.32             | 0.182         | -0.31             | 0.196         |
> > | rSLDS     | -0.36             | 0.128         | -0.29             | 0.231         |
> > | mp-rSLDS  | 0.06              | 0.804         | 0.04              | 0.886         |
> > | **CBP**   | **0.61**          | **0.005**     | **0.55**          | **0.014**     |
> >
> > In addition, we clarify that OT, HCP, and TLAE are all real fMRI datasets used to evaluate CBP’s reconstruction and task-variable prediction performance, whereas the Three-Task and Ten-Task Synthetic Datasets are controlled environments designed to analyze the modeling performance under idealized conditions.
> >
> > We have revised the manuscript to clearly distinguish the purposes of real-world and synthetic experiments and to include the missing baseline comparisons. These additions strengthen the empirical evaluation and more clearly demonstrate CBP’s practical utility. **(see revisions in Section 6 ⑤)**
> >
> > # **W4:**
> > We thank the reviewer for the careful observation. In the revised manuscript, we have conducted a systematic language review and formatting cleanup, including:
> >
> > (1) Correcting all grammar, capitalization, and typographical issues, including updating the section title to: "CBP Reveals Interpretable Cognitive Bases and Connectivity Patterns for Predicting Task-Related Variables."
> >
> > (2) Performing a thorough language polish and consistency check to ensure clearer, more precise writing aligned with machine learning and neuroscience publication standards.
> >
> > These revisions affect presentation only and do not change the technical content or conclusions. We appreciate the reviewer’s feedback, which has helped improve the clarity and overall quality of the manuscript.

---

> ### Comment · Reviewer_GVSN · 2025-11-27
>
> Thank you for the response. Authors have addressed my concerns and I have no further questions. I will raise my score from 4 to 6.

---

> > ### Author Response · Authors · 2025-11-28
> >
> > We sincerely thank the reviewer for the positive feedback and for taking the time to re-evaluate our submission. We greatly appreciate your decision to raise the score, and we are grateful for your insightful comments, which have helped us improve the clarity and overall quality of the manuscript.

---

### Official Review · Reviewer_VyUw · 2025-10-31

**Soundness:** 2
**Presentation:** 3
**Contribution:** 2
**Rating:** 2
**Confidence:** 4

**Summary:**

The paper introduces CBP (Cognitive Bases and connectivity Patterns), a dictionary-learning–based model designed to jointly capture shared cognitive components and their connectivity patterns across multiple fMRI tasks. Unlike existing foundation or state-space models—which are either hard to interpret or restricted to single tasks—CBP explicitly integrates cross-task information, modeling task-evoked brain dynamics as trajectories of a unified cognitive system. But the experiments are insufficient and the results were not very convincing.

**Strengths:**

1. The method is simple and easy to understand, and the defined losses are clear.
2. The paper demonstrates extensive experiments on synthetic, OT, HCP, and TLAE datasets. The qualitative analyses (e.g., identification of 12 cognitive bases and 8 connectivity motifs) show clear biological interpretability and align with canonical functional networks, which enhances the neuroscientific relevance of the work.

**Weaknesses:**

1. The performance is not state-of-the-art. This weakens the overall argument, as the proposed approach is less convincing when it does not outperform existing methods. And there is no statistical verification.
2. The comparison is rather limited; several widely used fMRI-based analysis methods were not considered (such as BrainGNN, BolT,BNT,Graphormer,NAGphormer...and foundation models BrainLM, BrainMass,Brain-JEPA......), which weakens the comprehensiveness of the evaluation.
3. The model introduces many regularization terms (Laplacian, sparsity, smoothness, decorrelation, etc.), but the intuition and relative contributions of each are not fully analyzed. The ablation table exists but lacks sufficient explanation and qualitative interpretation.
4. The paper is quite dense and mixes neuroscience motivation with technical formulation. The exposition could be streamlined to make the central idea—cross-task dictionary learning for cognitive bases—stand out more clearly.
5. While the integration across tasks is interesting, much of the mathematical formulation is reminiscent of existing decomposed LDS or sparse coding frameworks, with relatively incremental novelty at the algorithmic level.

**Questions:**

1. Are all datasets processed using the same MMP360 atlas for parcellation? Differences in atlas choice (e.g., MMP vs. Schaefer or Yeo) can strongly influence functional patterns and network structure, so clarification on this point would help interpret the cross-dataset results.
2. The paper identifies 12 “cognitive bases” that align with known networks. Can the authors provide quantitative validation (e.g., spatial correlation or network overlap with canonical atlases such as Yeo-17 or Power-264) to substantiate these qualitative findings?
3. How does the training complexity of CBP scale with the number of tasks and voxels? Given the high dimensionality of fMRI data, are there efficiency tricks (e.g., low-rank updates or GPU acceleration) that make the approach practical for larger datasets?

---

> ### Author Response · Authors · 2025-11-25
> **Part 1(VyUw: W1)**
>
> We thank the reviewer (VyUw) for the insightful comments. We have addressed the concerns regarding the **novelty, baseline comparisons, method, experiment details, quantitative evaluation of interpretability, and computational complexity**. Our response is organized into the following six parts.
>
> # Part 1(VyUw: W1)
>
> # **W1:**
> We thank the reviewer for the helpful suggestion. The main goal of our work is not to introduce another high-capacity deep learning model, but to develop an interpretable, non–deep-learning framework that can reveal scientifically meaningful cognitive bases and interaction patterns across tasks. **In many neuroscience applications, interpretability is essential rather than optional [1], whereas deep neural networks, as highly nonlinear black-box systems, often face challenges in providing reliable scientific insights**.
>
> Importantly, **despite not employing deep architectures, CBP achieves performance comparable to state-of-the-art deep models, and in several settings even surpasses them**. In the revised manuscript, we additionally include Wilcoxon signed-rank tests (see table below). These statistical results show that on datasets involving complex cognitive processing, CBP achieves significantly better performance than the strong deep architecture (BrainMoE [2]). To ensure fairness, the BrainMoE-style models were trained from scratch under the exact same data scale and task setting as CBP, without relying on any large-scale pretraining advantages.
>
> |**Metric**|**MSE(Y, Ŷ)**|||
> |-|-|-|-|
> |**Dataset**|**BrainMoE**|**CBP**|**p**|
> |Three-Task Synthetic|0.0051±0.0003|0.0052±0.0002|0.2324|
> |Ten-Task Synthetic|0.0030±0.0002|0.0021±0.0003|0.0020|
> |OT|0.0045±0.0002|0.0016±0.0003|0.0020|
> |HCP|0.0065±0.0018|0.0075±0.0004|0.1602|
> |TLAE|0.0088±0.0008|0.0023±0.0002|0.0020|
>
> |**Metric**|**ρ(Y, Ŷ)**|||
> |-|-|-|-|
> | **Dataset**|**BrainMoE**|**CBP**|**p**|
> |Three-Task Synthetic|0.9794±0.0110|0.9711±0.0002|0.0840|
> |Ten-Task Synthetic|0.9031±0.0014|0.9478±0.0003|0.0023|
> |OT|0.9059±0.0023|0.9116±0.0002|0.0016|
> |HCP|0.7804±0.0369|0.7617±0.0003|0.1055|
> |TLAE|0.8992±0.0006|0.9067±0.0002|0.0019|
>
>
> However, **achieving competitive predictive accuracy is not our ultimate goal. Our central objective is to identify interpretable cognitive bases and connectivity patterns, thereby providing transparent and scientifically meaningful insights for neuroscience applications**. **(see revisions in Section 1, Section 2, Section 6 ②, Appendix P)**
>
> [1] Noga Mudrik, Ryan Ly, Oliver Ruebel, and Adam S Charles. Creimbo: Cross-regional ensemble interactions in multi-view brain observations. The International Conference on Learning Representations, 2025.
>
> [2] Ziquan Wei, Tingting Dan, Tianlong Chen, and Guorong Wu. Brainmoe: Cognition joint embedding via mixture-of-expert towards robust brain foundation model. In The Thirty-ninth Annual Conference on Neural Information Processing Systems, 2025.

---

> > ### Author Response · Authors · 2025-11-25
> > **Part 2(VyUw: W2)**
> >
> > # Part 2(VyUw: W2)
> >
> > # **W2:**
> >
> > We thank the reviewer for the helpful suggestion. Our work aims to develop an interpretable, non–deep-learning framework for uncovering cognitive bases and interaction patterns that generalize across tasks. Accordingly, our original comparison focused on models that directly relate to our methodological and scientific goals:(1) interpretable non-deep dynamical models (SLDS, rSLDS, MP-rSLDS), and (2) representative deep dynamical models (LtrRNN, NetFormer, AMAG), covering recurrent, Transformer-based, and graph-based architectures. The comparison with deep models is intended to demonstrate that **even without deep architectures, CBP achieves modeling capability comparable to deep networks while providing interpretability that deep models typically lack**.
> >
> > To address the reviewer’s concern regarding the breadth of baselines, we further incorporated additional fMRI(EEG)-based analysis models (HGFM [1], BrainMoE [2]) in the revised manuscript. Under the exact same data scale and task setting as CBP, we trained HGFM-style and BrainMoE-style architectures from scratch, ensuring a fair comparison that does not rely on large-scale pretraining. The comparison results with the baselines are as follows:
> >
> >
> > |Dataset|Three-Task Synthetic Dataset||| Ten-Task Synthetic Dataset|||OT Dataset|HCP Dataset|TLAE Dataset|
> > |-|-|-|-|-|-|-|-|-|-|
> > |**Metric**|**MSE(Y, Ŷ)**|**MSE(A, Â)**|**MSE(X, X̂)**|**MSE(Y, Ŷ)**|**MSE(A, Â)**|**MSE(X, X̂)**|**MSE(Y, Ŷ)**|**MSE(Y, Ŷ)**|**MSE(Y, Ŷ)**|
> > |HGFM (2025)|0.0094±0.0014|/|/|0.0032±0.0008|/|/|0.0061±0.0010|0.0110±0.0005|0.0095±0.0006|
> > |BrainMoE (2025)|0.0051 ± 0.0003|/|/|0.0030±0.0002|/|/|0.0045±0.0002|0.0065±0.0018|0.0088±0.0008|
> > |CBP|0.0052±0.0002|0.0041±0.0003|0.0095±0.0002|0.0021±0.0003|0.0081±0.0002|0.0092±0.0002|0.0016±0.0003|0.0075±0.0004|0.0023±0.0002|
> >
> >
> > |Dataset|Three-Task Synthetic Dataset|||Ten-Task Synthetic Dataset|||OT Dataset|HCP Dataset|TLAE Dataset|
> > |-|-|-|-|-|-|-|-|-|-|
> > |**Metric**|**ρ(Y, Ŷ)**|**ρ(A, Â)**|**ρ(X, X̂)**|**ρ(Y, Ŷ)**|**ρ(A, Â)**|**ρ(X, X̂)**|**ρ(Y, Ŷ)**|**ρ(Y, Ŷ)**|**ρ(Y, Ŷ)**|
> > |HGFM (2025)|0.9227±0.0039|/|/|0.9085±0.0116|/|/|0.8634±0.0198|0.7207±0.0181|0.8751±0.0177|
> > |BrainMoE (2025)|0.9794±0.0110|/|/|0.9031±0.0014|/|/|0.9059±0.0023|0.7804±0.0369|0.8992±0.0006|
> > |CBP|0.9711±0.0002|0.9403±0.0003|0.9925±0.0002|0.9478±0.0003|0.8875±0.0004|0.9678±0.0003|0.9116±0.0002|0.7617±0.0003|0.9067±0.0002|
> >
> >
> > We also conducted statistical significance analysis, with the following results:
> >
> > |**Metric**|**MSE(Y, Ŷ)**|||
> > |-|-|-|-|
> > |**Dataset**|**BrainMoE**|**CBP**|**p**|
> > |Three-Task Synthetic|0.0051±0.0003|0.0052±0.0002|0.2324|
> > |Ten-Task Synthetic|0.0030±0.0002|0.0021±0.0003|0.0020|
> > |OT|0.0045±0.0002|0.0016±0.0003|0.0020|
> > |HCP|0.0065±0.0018|0.0075±0.0004|0.1602|
> > |TLAE|0.0088±0.0008|0.0023±0.0002|0.0020|
> >
> > |**Metric**|**ρ(Y, Ŷ)**|||
> > |-|-|-|-|
> > | **Dataset**|**BrainMoE**|**CBP**|**p**|
> > |Three-Task Synthetic|0.9794±0.0110|0.9711±0.0002|0.0840|
> > |Ten-Task Synthetic|0.9031±0.0014|0.9478±0.0003|0.0023|
> > |OT|0.9059±0.0023|0.9116±0.0002|0.0016|
> > |HCP|0.7804±0.0369|0.7617±0.0003|0.1055|
> > |TLAE|0.8992±0.0006|0.9067±0.0002|0.0019|
> >
> > These statistical results show that on datasets involving complex cognitive processing, CBP achieves significantly better performance than the strong deep architecture (BrainMoE).
> >
> > However, **achieving competitive predictive accuracy is not our ultimate goal. Our central objective is to identify interpretable cognitive bases and connectivity patterns, thereby providing transparent and scientifically meaningful insights for neuroscience applications**. **(see revisions in Section 1, Section 2, Section 6 ②, Appendix O, Appendix P)**
> >
> > [1] Xiangmin Han, Rundong Xue, Jingxi Feng, Yifan Feng, Shaoyi Du, Jun Shi, and Yue Gao. Hypergraph foundation model for brain disease diagnosis. IEEE Transactions on Neural Networks and Learning Systems, 36(10):17702–17716, 2025.
> >
> > [2] Ziquan Wei, Tingting Dan, Tianlong Chen, and Guorong Wu. Brainmoe: Cognition joint embedding via mixture-of-expert towards robust brain foundation model. In The Thirty-ninth Annual Conference on Neural Information Processing Systems, 2025.

---

> > > ### Author Response · Authors · 2025-11-25
> > > **Part 3(VyUw: W3, W4)**
> > >
> > > # Part 3(VyUw: W3, W4)
> > >
> > > # **W3:**
> > >
> > > We thank the reviewer for the helpful suggestion. The regularization terms in CBP are not heuristic add-ons, but are carefully designed to reflect the structural properties of fMRI data. Each term has a clear motivation and plays a distinct role in ensuring stability and interpretability. Specifically:
> > >
> > > (1) **Laplacian regularization: promoting spatial coherence and cross-task generalization.**
> > > This term encourages brain regions with similar temporal evolution patterns to be grouped into the same basis, forming coherent spatial clusters. It offers two key benefits: **(i) Suppressing local high-dimensional noise**, preventing each basis from becoming spatially fragmented; **(ii) Enhancing cross-task generalization**, encouraging the model to capture stable shared structures rather than overfitting to task-specific fluctuations. Ablation experiments show that removing this regularization term leads to noticeably noisier and less coherent bases.
> > >
> > > (2) **Sparsity regularization: promoting functional specificity.**
> > > Sparsity encourages each basis to focus on a limited subset of brain regions, forming meaningful and functionally specific subnetworks. Without sparsity, each basis tends to diffuse across the whole brain, losing functional specialization and reducing interpretability.
> > >
> > > (3) **Temporal smoothness regularization: enforcing physiologically realistic dynamics.**
> > > BOLD signals evolve slowly over time due to the hemodynamic response, and do not exhibit rapid, high-frequency oscillations. Temporal smoothness removes abrupt fluctuations and enforces temporally coherent, physiologically plausible interaction patterns. Ablation shows that without this term, the temporal patterns become unstable or noisy, deviating from realistic fMRI dynamics.
> > >
> > > (4) **Decorrelation regularization: preventing redundancy between patterns.**
> > > This term ensures that each pattern captures complementary structure instead of duplicating others. Removing it leads to highly redundant patterns and reduced interpretability.
> > >
> > > Taken together, the four regularizers constrain **spatial coherence, cross-task generalization, functional specificity, temporal stability, and non-redundancy among patterns**, each addressing a distinct property of fMRI data. Ablations confirm that their contributions are independent and non-interchangeable. Overall, they ensure the stability, generalizability, and spatial--temporal interpretability of CBP, demonstrating that the model design is principled rather than ad hoc. **(see revisions in Section 6 ②, Appendix R)**
> > >
> > > # **W4:**
> > >
> > > We thank the reviewer for the valuable suggestion. In the revised manuscript, we have streamlined and reorganized the exposition to more clearly highlight the central idea of "cross-task dictionary learning for extracting cognitive bases."  The main revisions are as follows:
> > >
> > > **We split the Section 3, "Problem Introduction and Approach," into two more focused subsections: "Problem Description" and "Hypothesis and Modeling Goals."** The former now exclusively presents the problem motivation and task background, while the latter clearly states the scientific hypotheses and high-level modeling goals. This separation prevents the mixing of motivation and technical detail, thereby improving overall conceptual clarity.
> > >
> > > **We added concise transition and summary sentences** at key points in the section to guide the reader and better connect the scientific motivation with the technical formulation.
> > >
> > > These revisions help improve readability and make the central idea of our approach more prominent and accessible.

---

> > > > ### Author Response · Authors · 2025-11-25
> > > > **Part 4(VyUw: W5, Q1)**
> > > >
> > > > # Part 4(VyUw: W5, Q1)
> > > >
> > > > # **W5:**
> > > >
> > > > We thank the reviewer for raising this important point. The novelty of CBP is reflected in its cross-task modeling mechanism, interpretability-driven design, and the scientific structures uncovered by this formulation, which are not accessible through existing LDS or sparse coding approaches.
> > > >
> > > > (1) **Cross-task modeling mechanism**: beyond the capacity of LDS/sparse coding.
> > > > CBP introduces a representational structure that simultaneously captures: (i) shared cognitive bases, (ii) shared connectivity patterns, and (iii) task-specific temporal trajectories. This design integrates cross-task invariances with condition-dependent modulations, which is not supported by standard LDS (single dynamical system) nor sparse coding (absence of dynamics). This constitutes a new
> > > > modeling paradigm rather than an incremental variation.
> > > >
> > > > (2) **Interpretability-driven algorithmic design**.
> > > > The Laplacian, sparsity, temporal smoothness, and decorrelation regularizers are not heuristic add-ons. They are carefully designed to correspond to the spatial, functional and temporal of fMRI data. Ablations demonstrate that each term is necessary for obtaining stable, non-redundant, and neurobiologically interpretable bases and patterns, which are behaviors not exhibited by LDS or sparse coding.
> > > >
> > > > (3) **Scientific contribution**: discovering cross-task cognitive bases and connectivity patterns.
> > > > Existing LDS/rSLDS or sparse coding methods operate on single-task settings and cannot recover structure that is shared across heterogeneous cognitive tasks. In contrast, CBP enables:
> > > >
> > > > (i) identification of **stable and reproducible cognitive bases and connectivity patterns across 103 tasks**;
> > > >
> > > > (ii) analysis of how these bases and connectivity patterns are reused across tasks and time;
> > > >
> > > > (iii) new neuroscientific insights into the task-invariant components of brain activity and their ability to generalize to new task conditions.
> > > >
> > > > These findings are not obtainable from existing models.
> > > >
> > > > Overall, while CBP uses standard optimization tools, its **cross-task modeling capabilities, interpretability-oriented regularization  scientific discovery, and representational structure** are fundamentally different from existing approaches. These elements together form a new conceptual framework rather than an incremental algorithmic modification.
> > > >
> > > > # **Q1:**
> > > >
> > > > We thank the reviewer for the helpful question. All datasets used in our study were processed **using the same MMP360 atlas** for parcellation, ensuring that cross-dataset results are fully comparable and not confounded by differences in atlas choice.

---

> > > > > ### Author Response · Authors · 2025-11-25
> > > > > **Part 5(VyUw: Q2)**
> > > > >
> > > > > # Part 5(VyUw: Q2)
> > > > >
> > > > > # **Q2:**
> > > > >
> > > > > We thank the reviewer for raising this important question. To ensure objectivity and reproducibility, we introduce three quantitative interpretability metrics that establish a principled mapping between the learned bases and the canonical Yeo-7 functional networks (**Yeo-17: Section 6 ④**). These metrics evaluate (i) spatial alignment, (ii) functional specificity, and (iii) global coverage of large-scale networks.
> > > > >
> > > > > (1) Basis--Network Dice Overlap (alignment).
> > > > > This metric quantifies how strongly a learned basis overlaps with each Yeo-7 functional network, thus replacing qualitative matching with a rigorous numerical measure. For the $b$-th basis, $A^{b} = (A_{1,b}, \dots, A_{N,b})$ denotes its weights across cortical ROIs, where $N = 360$. We first compute an activation threshold that selects its top $15\%$ highest-weighted
> > > > > ROIs. This threshold is given by the 85th percentile of its weight vector: $\theta_b = \mathrm{Percentile}(A^{(b)}, 85)$.
> > > > >
> > > > > We then identify the top-15\% activation set of basis $b$ as:
> > > > > \begin{equation}
> > > > >     S_b = ( i \mid A_{i,b} \ge \theta_b ).
> > > > > \end{equation}
> > > > > Let $R_k$ denote the ROI set of the $e$-th Yeo-7 network.
> > > > > The Dice overlap is
> > > > > \begin{equation}
> > > > >     D_{b,e} =
> > > > > \frac{2 |S_b \cap R_e|}{|S_b| + |R_e|},
> > > > > \end{equation}
> > > > > where $e= 1,...,E$, $E = 7$. A higher Dice value indicates stronger alignment between basis $b$ and network $e$. As shown in the table below, by computing the Dice overlap between each basis and the Yeo-7 functional networks, we obtain a basis–network mapping matrix, revealing strong correspondence between OT-derived bases and canonical networks.
> > > > >
> > > > > |Basis|Visual|Somatomotor|Dorsal Attention|Ventral Attention|Limbic|Frontoparietal|Default|
> > > > > |-|-|-|-|-|-|-|-|
> > > > > |**A1**|0.12|**0.21**|0.06|0.02|**0.51**|0.06|0.12|
> > > > > |**A2**|0.09|0.06|0.00|0.06|**0.34**|0.08|**0.37**|
> > > > > |**A3**|0.00|0.04|0.14|0.18|0.02|**0.36**|**0.25**|
> > > > > |**A4**|0.00|0.19|0.02|**0.31**|0.00|**0.46**|0.06|
> > > > > |**A5**|0.04|**0.26**|0.16|0.16|**0.36**|0.02|0.09|
> > > > > |**A6**|**0.48**|0.11|**0.16**|0.02|0.10|0.02|0.10|
> > > > > |**A7**|**0.23**|0.17|**0.45**|0.02|0.02|0.10|0.04|
> > > > > |**A8**|**0.76**|0.00|**0.18**|0.02|0.00|0.02|0.00|
> > > > > |**A9**|0.09|0.06|0.02|0.10|**0.24**|0.06|**0.40**|
> > > > > |**A10**|0.05|0.15|0.06|**0.18**|0.00|0.02|**0.44**|
> > > > > |**A11**|0.04|**0.22**|0.08|0.04|0.05|**0.38**|0.19|
> > > > > |**A12**|0.04|**0.52**|0.00|**0.31**|0.02|0.02|0.09|
> > > > >
> > > > >
> > > > > (2) Basis Concentration (functional specificity).
> > > > > This metric determines whether a basis is specialized (localized in one or a few networks) or
> > > > > integrative (distributed across multiple networks). We first normalize the Dice scores:
> > > > > \begin{equation}
> > > > >     p_{b,e} =
> > > > > \frac{D_{b,e}}{
> > > > > \sum_{e'} D_{b,e'}}.
> > > > > \end{equation}
> > > > > and then compute the Shannon entropy:
> > > > > \begin{equation}
> > > > >     H_b = - \sum_{e} p_{b,e}\log p_{b,e},
> > > > > \end{equation}
> > > > > The concentration index is defined as:
> > > > > \begin{equation}
> > > > >     C_b = 1 - \frac{H_b}{\log E}.
> > > > > \end{equation}
> > > > > The overall concentration is then defined as:
> > > > > \begin{equation}
> > > > >     \mathrm{Concentration}
> > > > > = \frac{1}{K}\sum_{b=1}^{K} C_b.
> > > > > \end{equation}
> > > > > where $b= 1,...,K$, $K = 12$. High Concentration reflects strong functional specificity.
> > > > >
> > > > > (3) Functional Network Coverage (global completeness).
> > > > > This metric assesses whether the full set of bases collectively covers all major functional
> > > > > systems, preventing collapse into a few high-variance networks. For each Yeo-7 network $e$, we compute the maximum normalized Dice:
> > > > > \begin{equation}
> > > > >     m_e =
> > > > > \max_{b} p_{b,e}.
> > > > > \end{equation}
> > > > > A network is considered sufficiently represented when this value exceeds a predefined threshold: $\delta_e = (1, (m_e \ge \tau); 0, (m_e < \tau))$
> > > > >
> > > > > The overall coverage is then defined as:
> > > > > \begin{equation}
> > > > >     \mathrm{Coverage}
> > > > > = \frac{1}{E}\sum_{e=1}^{E} \delta_e.
> > > > > \end{equation}
> > > > > Higher coverage indicates that the learned bases collectively capture the full spectrum of large-scale functional networks, which reflects the completeness and robustness of the bases.
> > > > >
> > > > > To further validate the interpretability of the bases learned from OT ($A_{\mathrm{OT}}$), we compared them with the bases trained directly on HCP ($A_{\mathrm{HCP}}$) using our concentration and coverage metrics. The results show that the OT-derived bases ($A_{\mathrm{OT}}$) exhibit higher concentration and much higher coverage, indicating substantially stronger neurobiological interpretability (see table below). This is likely because HCP includes fewer tasks and has uneven task durations, limiting the richness of cognitive sampling required to learn well-structured bases.
> > > > >
> > > > > |Metric|Concentration↑|Coverage↑|
> > > > > |-|-|-|
> > > > > |A(HCP)|0.1367|0.5714|
> > > > > |A(OT)|**0.2727**|**1.0**|
> > > > >
> > > > > In summary, the learned bases quantitatively align with canonical functional systems, exhibit meaningful specificity, and achieve broad functional coverage. This grounding ensures that both the bases and the derived connectivity patterns are neuroscientifically interpretable in a principled and reproducible manner. **(see revisions in Section 6 ④, Appendix N)**

---

> > > > > > ### Author Response · Authors · 2025-11-25
> > > > > > **Part 6(VyUw: Q3)**
> > > > > >
> > > > > > # Part 6(VyUw: Q3)
> > > > > >
> > > > > > # **Q3:**
> > > > > >
> > > > > > We thank the reviewer for the helpful question regarding computational complexity. CBP is a non-deep-learning, region-level model operating on ROI time series (MMP360) rather than voxel-wise data. Thus, the computational dimensionality is determined by the number of ROIs ($N=360$), which is orders of magnitude smaller than voxel space and substantially reduces overall cost.
> > > > > >
> > > > > > Let $S$ be the number of subjects, $M$ be the number of tasks, and $T _ m^{(s)}$ be the length of task $m$ for subject $s$. Let $T _ {\mathrm{total}} = \sum _ {s=1}^S \sum _ {m=1}^M T _ m^{(s)}$ denote the total number of time points across all subject–task pairs, and let $K$ and $P$ denote the number of cognitive bases and connectivity patterns, respectively. One alternating-optimization iteration of CBP consists of three updates:
> > > > > >
> > > > > > (i) Cognitive bases update ($A$): updating $A$ involves the reconstruction term $\|Y-AX\| _ F^2$ and the Laplacian regularizer, with time complexity $O (N K T _ {\mathrm{total}})$;
> > > > > >
> > > > > > (ii) Latent states and dynamic coefficients update ($X$ and $C$): this step is dominated by the dynamical consistency term and has complexity $O \big(T _ {\mathrm{total}} (N K + P K^2)\big)$;
> > > > > >
> > > > > > (iii) Connectivity patterns update ($\{f _ p\} _ {p=1}^P$): this step also scales as $O \big(T _ {\mathrm{total}} P K^2\big)$.
> > > > > >
> > > > > > **Combining these terms, the per-iteration complexity is $O \big(T _ {\mathrm{total}} (N K + P K^2)\big)$, which is linear in the total number of time points $T _ {\mathrm{total}}$ (and thus linear in the number of tasks and subjects). Since $K$ and $P$ are small in practice (e.g., $K=12$, $P=8$) and $N=360$ is fixed by the atlas, the overall training cost scales well to larger multi-task datasets.**
> > > > > >
> > > > > > All updates rely on **vectorized matrix operations**, making CBP efficient on CPU and naturally compatible with GPU acceleration. In our experiments, training on the largest dataset fits comfortably on a single workstation; no additional tricks such as low-rank approximations are required. We have included a concise complexity discussion in the revised manuscript **(see revisions in Appendix C)**.

---

> > > > > > > ### Comment · Reviewer_VyUw · 2025-11-27
> > > > > > >
> > > > > > > I appreciate the author's efforts during the rebuttal period, and most of my concerns have been resolved. However, I still have one more concern, which is "The statement 'CBP significantly outperforms BrainMoE' selectively emphasizes only the datasets where CBP performances are better and ignore datasets where BrainMoE is numerically superior (Table 9). However, the differences in those datasets are not statistically significant, and the overall conclusion may appear as an over statement without explicit restrictions to the datasets with significant differences. BTW, I'd like to raise my score. Thanks.

---

> > > > > > > > ### Author Response · Authors · 2025-11-28
> > > > > > > >
> > > > > > > > We sincerely thank the reviewer for this thoughtful clarification. We fully agree with the concern regarding potential overstatement. Our intention was to highlight improvements that are statistically significant, rather than emphasize numerical differences alone.
> > > > > > > >
> > > > > > > > To avoid any ambiguity, we have revised the statement in the manuscript to:
> > > > > > > >
> > > > > > > > **“Statistical significance analyses (Wilcoxon signed-rank tests) show that CBP outperforms BrainMoE on datasets where the difference is statistically significant, and performs comparably on datasets where the numerical differences are not significant.” (see revisions in Section 6 ②, and Appendix P)**
> > > > > > > >
> > > > > > > > We appreciate the reviewer for bringing this point to our attention, as this clarification ensures that our conclusions remain strictly aligned with statistical evidence. We are also grateful for the reviewer’s willingness to raise the score, and we hope that the revisions fully address this final concern. **If the reviewer has any remaining comments or suggestions, we would be grateful to receive them and will make every effort to further strengthen the paper.**

---

### Official Review · Reviewer_tXJt · 2025-11-01

**Soundness:** 3
**Presentation:** 2
**Contribution:** 3
**Rating:** 6
**Confidence:** 3

**Summary:**

This paper proposes CBP that learns a shared cognitive basis space and a set of connectivity patterns that describe how these bases interact over time. The model is designed to be both interpretable and generalizable across many tasks, and the authors provide experiments on synthetic data, OT fMRI (over 100 tasks), HCP tasks.

**Strengths:**

1. The work explicitly separates and models different components of brain activity (shared bases, task-specific activation trajectories, and connectivity patterns). This gives the model a high level of interpretability.

2. The experimental design is rich and logically structured. The authors move from controlled synthetic data, to large-scale multi-task fMRI, and then to HCP (task decoding).

3. The model provides intuitive visualizations. The learned bases are shown as cortical maps that appear aligned with known functional systems.

**Weaknesses:**

1. The paper assumes that different tasks are different trajectories in the same cognitive system, i.e. that a single shared set of bases and patterns can explain many (or all) tasks. Is this really sufficient? Are the learned bases universal, or are they tied to a specific dataset and experimental protocol? What happens if we change the acquisition condition for the same subject (for example, different scanner runs, different cognitive state, different instructions)? I suggest adding a sensitivity analysis that tests how stable the learned bases are under such perturbations.

2. The paper interprets $A$ as functional subnetworks. Is this mapping predefined in any way, or purely learned in a data-driven way? The paper suggests that using a larger number of bases improves reconstruction and downstream prediction. If so, how should we interpret different choices of the number of bases? Right now $A$ feels partly like a posterior observation, which weakens the claim of intrinsic interpretability.

3. The model explains temporal evolution of brain activity as a combination of multiple linear interaction patterns between bases. Real neural dynamics are often strongly nonlinear and state-dependent. The paper should discuss how well this linear-by-parts approximation can capture nonlinear behavior. For example, under what conditions would this approximation fail?

4. The transfer experiment mainly goes in one direction, from very rich multi-task dataset OT to a fewer-task dataset HCP. If we had an even more fine-grained cognitive taxonomy than OT (more subtle task distinctions), would the bases learned on OT still work well?
The paper should also report a baseline where the model is trained directly on HCP, and then compare that to the transferred version from OT. How different are the learned bases and patterns in these two cases?

**Questions:**

How exactly are the learned bases $A$ linked to known functional subnetworks? Is this a qualitative, visual matching (this looks like auditory cortex, so we call it auditory)? Similarly, how are the learned connectivity patterns mapped to cognitive labels like “perception–action loop” or “integration pathway”?

---

> ### Author Response · Authors · 2025-11-25
> **Part 1(tXJt: W1, W2)**
>
> We thank the reviewer (tXJt) for the insightful comments. We have addressed the concerns regarding the **validity of the CBP model, the reliability of the discovered cognitive bases, and the quantitative evaluation of interpretability**. Our response is organized into the following three parts.
>
> # Part 1(tXJt: W1, W2)
>
> **W1:**
> We thank the reviewer for the insightful suggestion. To assess whether the learned bases depend on specific preprocessing or acquisition choices, we performed two sensitivity analyses that explicitly measure the similarity between the original bases A and the bases Â learned from perturbed datasets.
>
> (1) **Temporal-resolution perturbation:**
>
> We simulated changes in TR by upsampling the fMRI time axis with interpolation scales from 1.5 to 4.0, retrained CBP on each perturbed dataset, and computed the similarity ρ(A, Â). The bases remained highly consistent with the originals:
> | Interpolation scale | 1.5           | 2.0           | 2.5           | 3.0           | 3.5           | 4.0           |
> |---------------------|---------------|---------------|---------------|---------------|---------------|---------------|
> | ρ(A, Â) ↑           | 0.9074±0.0001 | 0.8952±0.0001 | 0.9348±0.0001 | 0.8845±0.0001 | 0.8948±0.0001 | 0.8960±0.0001 |
>
> These results show that CBP is robust to reasonable variations in TR and preprocessing.
>
> (2) **Noise perturbation:**
>
> We progressively added Gaussian noise with increasing variance to the fMRI time series, retrained CBP, and again measured ρ(A, Â). The bases remained highly similar under low-to-moderate noise levels:
>
> | σ_noise / σ_data | $10^{-6}$ | $10^{-5}$ | $10^{-4}$ | $10^{-3}$ | $10^{-2}$ | $10^{-1}$ | $10^{0}$ | $10^{1}$ |
> |------------------|-----------|-----------|-----------|-----------|-----------|-----------|----------|----------|
> | ρ(A, Â) ↑        | 0.9999 | 0.9999 | 0.9999| 0.9999 | 0.9999| 0.9991 | 0.9301| 0.1354 |
> |       | ±0.0002 | ±0.0001 | ±0.0001 | ±0.0001 | ±0.0001 | ±0.0004 | ±0.0211 | ±0.0060 |
>
> This demonstrates that the learned bases are robust to realistic variations in signal quality.
>
> Together, these analyses demonstrate that the learned cognitive bases are not tied to a particular preprocessing pipeline or acquisition condition. Instead, they represent stable, reproducible structures that persist under substantial perturbations in temporal resolution and noise level. **(see revisions in Section 6 ④)**
>
> **W2:**
> We thank the reviewer for raising this important point.
>
> (1) **The bases $A$ are not predefined in any way.**
>
> They are learned fully in a data-driven manner through joint optimization across all tasks, without imposing any atlas, template, or prior network structure.
>
> (2) **The interpretability of $A$ is intrinsic.**
>
> Each column of $A$ functions as a spatial functional subnetwork that linearly composes to reconstruct task-evoked activity across all tasks. CBP explicitly enforces a decomposition into shared cognitive bases $A$ and latent task trajectories $X$, ensuring that each basis corresponds to a reproducible large-scale network rather than a post hoc visualization artifact.
>
> (3) **On the choice of the number of bases $K$.**
>
> Following the work [1], we systematically evaluated $K$ from 2 to 40. As shown in the table below, explained variance and reconstruction accuracy plateau around $K=12$, and larger values yield only marginal improvements. We therefore adopt the elbow criterion and set $K=12$, which provides a good balance between model complexity and interpretability.
>
> |       K             | 2      | 4      | 6      | 8      | 10     | 12     |14      |16      |18      |20      |30      |40      |
> |---------------------|--------|--------|--------|--------|--------|--------|--------|--------|--------|--------|--------|--------|
> | Explained Variance ↑| 0.3483 | 0.4453 | 0.5134 | 0.5609 | 0.5786 | 0.6126 | 0.6304 | 0.6398 | 0.6435 | 0.6546 | 0.6610 | 0.6737 |
>
>
> |       K             | 2      | 4      | 6      | 8      | 10     | 12     |14      |16      |18      |20      |30      |40      |
> |---------------------|--------|--------|--------|--------|--------|--------|--------|--------|--------|--------|--------|--------|
> | MSE ↓               | 0.0024 | 0.0020 | 0.0018 | 0.0016 | 0.0015 | 0.0014 | 0.0014 | 0.0013 | 0.0013 | 0.0013 | 0.0012 | 0.0012 |
>
> Taken together, these results show that the learned bases are fully data-driven and robust to the choice of $K$, demonstrating that their interpretability arises from the model’s design rather than from posterior inspection. **(see revisions in Section 6 ③)**
>
> [1] Camden J MacDowell and Timothy J Buschman. Low-dimensional spatiotemporal dynamics underlie cortex-wide neural activity. Current Biology, 30(14):2665–2680, 2020

---

> ### Author Response · Authors · 2025-11-25
> **Part 2(tXJt: W3, W4)**
>
> # Part 2(tXJt: W3, W4)
>
> **W3:**
> We thank the reviewer for the insightful suggestion. To evaluate how well the proposed linear-by-parts formulation can capture nonlinear neural dynamics, we use a controlled synthetic system where the
> nonlinearity can be adjusted explicitly:
> \begin{equation}
> y = x^{\tau},
> \end{equation}
> where the input x is sampled as 400 evenly spaced points over the interval [−2, 2]; the exponent $\tau$ directly controls the degree of nonlinearity. This system is widely used in nonlinear system analysis because: (1) it preserves input variance and does not collapse to a fixed point; (2) it allows continuous tuning of nonlinearity strength; (3) it provides a clean testbed to evaluate piecewise-linear approximators.
>
> (1) **Performance under increasing nonlinearity.**
> As shown in the table below (number of bases $K=2$), CBP maintains high reconstruction accuracy across a wide range of nonlinearity levels. Its performance remains stable even for the strongly nonlinear level ($\tau=6$), and only starts to degrade for very large exponents where the input–output relation becomes extremely steep and highly nonlinear.
> |**Nonlinear level $\tau$**|**1**|**2**|**3**|**4**|**5**|**6**|**7**|**8**|
> |-|-|-|-|-|-|-|-|-|
> |**ρ(Y, Ŷ)↑**|1.0000±0.0001|1.0000±0.0001|0.9999±0.0001|0.9959±0.0006|0.9626±0.0014|0.8708±0.0015|0.8190±0.0029|0.6658±0.0029|
>
> (2) **Effect of increasing the number of bases.**
> Moreover, when the nonlinearity is extremely strong, increasing the number of bases $K$ substantially improves reconstruction accuracy, consistent with the intuition that a larger basis set provides a finer piecewise-linear approximation:
> |**Number of bases $K$**|**2**| **4**|**6**|**8**|**10**|**12**|**14**|**16**|**18**|**20**|
> |-|-|-|-|-|-|-|-|-|-|-|
> |**ρ(Y, Ŷ)↑**|0.6658±0.0029|0.7705±0.0021|0.8511±0.0015|0.8943±0.0014|0.9289±0.0011|0.9523±0.0007|0.9698±0.0006|0.9799±0.0006|0.9884±0.0004|0.9949±0.0003|
>
> These analyses support the view that CBP behaves as an effective piecewise-linear approximator whose expressiveness increases with the number of bases, enabling it to model nonlinear dynamics over a broad operating range. **(see revisions in Section 6 ④)**
>
> **W4:**
> We thank the reviewer for this insightful comment. To systematically examine the dependence of the learned cognitive bases on cognitive-task granularity, and to compare "training directly on HCP" versus "transferring from OT," we conducted two additional analyses.
>
> (1) **Reliability under different task sampling scales.** We randomly sample different numbers of cognitive tasks from the OT dataset and learn the bases independently. As shown in the table below, the results show that as the number of tasks increases, the learned cognitive bases become stable and exhibit consistent interpretability. This indicates that the bases do not rely on any specific task protocol and remain robust once task sampling reaches a sufficient scale.
>
> |**Proportion of tasks sampled**|**0.2**|**0.3**|**0.4**|**0.5**|**0.6**|**0.7**|**0.8**|**0.9**|**1.0**|
> |-|-|-|-|-|-|-|-|-|-|
> |**ρ(Y, Ŷ)↑**|0.6070±0.0470|0.6574±0.0643|0.6942±0.0080|0.6988±0.0370|0.8763±0.0384|0.8302±0.0187|0.8937±0.0225|0.8579±0.0137|0.9039±0.0150|
>
> (2) **Comparison between bases trained on HCP and those transferred from OT.**
> We trained CBP directly on HCP to obtain a set of bases $A_{\mathrm{HCP}}$, and compared them with the OT-learned bases $A_{\mathrm{OT}}$. The results show that the OT-derived bases ($A_{\mathrm{OT}}$) exhibit higher concentration and much higher coverage, indicating substantially stronger neurobiological interpretability (see table below). This is likely because HCP includes fewer tasks and has imbalanced task durations, limiting the richness of cognitive sampling required to learn well-structured bases. (The concentration and coverage metrics are interpretability indicators proposed in our work; definitions are provided in the Appendix N and in our response to Q1.)
>
> |Metric|Concentration↑|Coverage↑|
> |-|-|-|
> |A(HCP)|0.1367|0.5714|
> |A(OT)|**0.2727**|**1.0**|
>
>
> Taken together, these analyses demonstrate that: (1) with sufficiently rich cognitive sampling, CBP learns stable, reproducible, and interpretable cognitive bases; (2) directly training on the limited-task HCP dataset does not produce bases of comparable quality. We have included these new experiments and the corresponding figures in the revised manuscript **(see revisions in Section 6 ④, Appendix N)**.

---

> > ### Author Response · Authors · 2025-11-25
> > **Part 3(tXJt: Q1)**
> >
> > # Part 3(tXJt: Q1)
> >
> > **Q1:**
> > We thank the reviewer for raising this important question. To ensure objectivity and reproducibility, we introduce three quantitative interpretability metrics that establish a principled mapping between the learned bases and the canonical Yeo-7 functional networks (**Yeo-17: Section 6 ④**). These metrics evaluate (i) spatial alignment, (ii) functional specificity, and (iii) global coverage of large-scale networks.
> >
> > (1) **Basis--Network Dice Overlap (alignment)**.
> > This metric quantifies how strongly a learned basis overlaps with each Yeo-7 functional network,
> > thus replacing qualitative matching with a rigorous numerical measure.
> > For the $b$-th basis, $A^{b} = (A_{1,b}, \dots, A_{N,b})$ denotes its weights across cortical ROIs, where $N = 360$. We first compute an activation threshold that selects its top $15\%$ highest-weighted
> > ROIs. This threshold is given by the 85th percentile of its weight vector:
> > \begin{equation}
> >     \theta_b = \mathrm{Percentile}(A^{(b)}, 85).
> > \end{equation}
> > We then identify the top-15\% activation set of basis $b$ as:
> > \begin{equation}
> >     S_b = ( i \mid A_{i,b} \ge \theta_b ).
> > \end{equation}
> > Let $R_k$ denote the ROI set of the $e$-th Yeo-7 network.
> > The Dice overlap is
> > \begin{equation}
> >     D_{b,e} =
> > \frac{2 |S_b \cap R_e|}{|S_b| + |R_e|},
> > \end{equation}
> > where $e= 1,...,E$, $E = 7$. A higher Dice value indicates stronger alignment between basis $b$ and network $e$. As shown in the table below, by computing the Dice overlap between each basis and the Yeo-7 functional networks, we obtain a basis–network mapping matrix, revealing strong correspondence between OT-derived bases and canonical networks.
> >
> > |Basis|Visual|Somatomotor|Dorsal Attention|Ventral Attention|Limbic|Frontoparietal|Default|
> > |-|-|-|-|-|-|-|-|
> > |**A1**|0.12|**0.21**|0.06|0.02|**0.51**|0.06|0.12|
> > |**A2**|0.09|0.06|0.00|0.06|**0.34**|0.08|**0.37**|
> > |**A3**|0.00|0.04|0.14|0.18|0.02|**0.36**|**0.25**|
> > |**A4**|0.00|0.19|0.02|**0.31**|0.00|**0.46**|0.06|
> > |**A5**|0.04|**0.26**|0.16|0.16|**0.36**|0.02|0.09|
> > |**A6**|**0.48**|0.11|**0.16**|0.02|0.10|0.02|0.10|
> > |**A7**|**0.23**|0.17|**0.45**|0.02|0.02|0.10|0.04|
> > |**A8**|**0.76**|0.00|**0.18**|0.02|0.00|0.02|0.00|
> > |**A9**|0.09|0.06|0.02|0.10|**0.24**|0.06|**0.40**|
> > |**A10**|0.05|0.15|0.06|**0.18**|0.00|0.02|**0.44**|
> > |**A11**|0.04|**0.22**|0.08|0.04|0.05|**0.38**|0.19|
> > |**A12**|0.04|**0.52**|0.00|**0.31**|0.02|0.02|0.09|
> >
> > (2) **Basis Concentration (functional specificity)**.
> > This metric determines whether a basis is specialized (localized in one or a few networks) or
> > integrative (distributed across multiple networks). We first normalize the Dice scores:
> > \begin{equation}
> >     p_{b,e} =
> > \frac{D_{b,e}}{
> > \sum_{e'} D_{b,e'}}.
> > \end{equation}
> > and then compute the Shannon entropy:
> > \begin{equation}
> >     H_b = - \sum_{e} p_{b,e}\log p_{b,e},
> > \end{equation}
> > The concentration index is defined as:
> > \begin{equation}
> >     C_b = 1 - \frac{H_b}{\log E}.
> > \end{equation}
> > The overall concentration is then defined as:
> > \begin{equation}
> >     \mathrm{Concentration}
> > = \frac{1}{K}\sum_{b=1}^{K} C_b.
> > \end{equation}
> > where $b= 1,...,K$, $K = 12$. High Concentration strong functional specificity. As shown in our response to W4, we find that the bases learned from OT exhibit higher concentration compared to the bases trained directly on HCP.
> >
> > |Metric|Concentration↑|
> > |-|-|
> > |A(HCP)|0.1367|
> > |A(OT)|**0.2727**|
> >
> >
> > (3) **Functional Network Coverage (global completeness)**.
> > This metric assesses whether the full set of bases collectively covers all major functional
> > systems, preventing collapse into a few high-variance networks. For each Yeo-7 network $e$, we compute the maximum normalized Dice:
> > \begin{equation}
> >     m_e =
> > \max_{b} p_{b,e}.
> > \end{equation}
> > A network is considered sufficiently represented when this value exceeds a predefined threshold: $\delta_e = (1, (m_e \ge \tau); 0, (m_e < \tau))$.
> >
> > The overall coverage is then defined as:
> > \begin{equation}
> >     \mathrm{Coverage}
> > = \frac{1}{E}\sum_{e=1}^{E} \delta_e.
> > \end{equation}
> > Higher coverage indicates that the learned bases collectively capture the full spectrum of large-scale functional networks, which reflects the completeness and robustness of the bases. As shown in our response to W4, we find that the bases learned from OT exhibit higher coverage compared to the bases trained directly on HCP.
> >
> > |Metric|Coverage↑|
> > |-|-|
> > |A(HCP)|0.5714|
> > |A(OT)|**1.0**|
> >
> > In summary, the learned bases quantitatively align with canonical functional systems, exhibit meaningful specificity, and achieve broad functional coverage. This grounding ensures that both the bases and the derived connectivity patterns are neuroscientifically interpretable in a principled and reproducible manner. **(see revisions in Section 6 ④, Appendix N)**

---

### Author Response · Authors · 2025-12-02
**Global Response (1/2)**

# Global Response (1/2)

We sincerely thank all reviewers and the ACs for their thoughtful evaluation of our submission. We greatly appreciate the time and effort invested in reviewing the manuscript and providing positive feedback. We are pleased that all reviewers acknowledged the importance of our scientific finding that human brain activity exhibits a set of task-invariant cognitive bases and connectivity patterns, as well as the high interpretability of the proposed model. **We are also glad that the concerns raised by Reviewers VyUw and GVSN have been fully resolved, and we sincerely appreciate their decision to raise their scores.**

Our contributions can be summarized in the following three key points:

(1) Methodology: We present CBP, an interpretable, non–deep-learning framework for cross-task brain modeling. It exploits shared information across tasks and accurately recovers latent components while maintaining state-of-the-art performance.

(2) **Scientific Discovery: CBP uncovers a stable set of cognitive bases and connectivity patterns in the human brain. The reliability of these discoveries is supported by extensive quantitative evaluations and a battery of perturbation analyses, demonstrating their robustness.**

(3) Generalization: CBP, grounded in shared cognitive components, generalizes well to the HCP and multi-stage learning datasets. It accurately predicts task states and learning outcomes and reveals the key cognitive connectivity patterns that drive these behaviors.

To further assist the reviewers and ACs in evaluating our submission, we summarize the concerns raised by the reviewers and our corresponding revisions across five key areas:

### **(1) Reliability of the scientific findings (cognitive bases and connectivity patterns). (Reviewers tXJt, pNLA)**

(i) Temporal-resolution perturbation.
We added experiments that systematically vary the temporal resolution (simulating changes in TR by interpolating the fMRI time axis with different upsampling factors). Results show that the learned cognitive bases remain highly robust under reasonable changes in temporal resolution and preprocessing. **(see revisions in Section 6 ④)**

(ii) Noise perturbation.
We added experiments with varying noise levels. The cognitive bases remain stable across practical variations in signal quality. **(see revisions in Section 6 ④)**

(iii) Different task-sampling scales.
We added experiments that vary the amount of cognitive-task sampling. Results show that the cognitive bases do not depend on any specific task protocol and become highly stable once task sampling reaches a sufficient scale. **(see revisions in Section 6 ④)**

(iv) Choice of the numbers of bases and patterns.
Following the work [1], we provide principled procedures for selecting the numbers of cognitive bases and connectivity patterns. The results show that the learned cognitive bases and connectivity patterns are fully data-driven and robust to the choice of K. **(see revisions in Section 6 ③)**

### **(2) Quantitative evaluation of the interpretability of cognitive bases. (Reviewers tXJt, VyUw)**

We introduced three quantitative interpretability metrics (alignment, concentration, and coverage) and added corresponding experiments. Results show that the learned cognitive bases exhibit strong neuroscientific interpretability and align well with canonical functional networks (Yeo-7 / Yeo-17). **(see revisions in Section 6 ④, Appendix N)**

### **(3) Evaluation of modeling performance. (Reviewers tXJt, VyUw, pNLA)**

(i) Additional deep-learning baselines and statistical tests.
We added comparisons with new deep baselines (HGFM [2], BrainMoE [3]) and performed Wilcoxon signed-rank tests. Results show that CBP significantly outperforms BrainMoE on datasets where differences are statistically significant and performs comparably on datasets where numerical differences are not significant. **The comparison with deep models is intended to show that even without deep learning architectures, CBP can achieve reconstruction performance comparable to deep networks while providing the interpretability that deep models typically lack. However, achieving competitive predictive accuracy is not our ultimate goal. Our central objective is to identify interpretable cognitive bases and connectivity patterns, thereby providing transparent and scientifically meaningful insights for neuroscience applications. In neuroscience-oriented applications, interpretability is essential rather than optional [4]. (see revisions in Section 1, Section 2, Section 6 ②, Appendix O, Appendix P)**

(ii) Nonlinearity analysis.
We added a controlled synthetic system with tunable nonlinearity to further evaluate CBP. Results show that CBP can model nonlinear dynamics across a broad operating range.**(see revisions in Section 6 ④, Appendix M)**

(iii) Complexity analysis.
We added computational complexity analysis, demonstrating that CBP is efficient. **(see revisions in Appendix C)**

---

> ### Author Response · Authors · 2025-12-02
> **Global Response (2/2)**
>
> # Global Response (2/2)
>
> ### **(4) Performance evaluation for task-related variable prediction. (Reviewer GVSN)**
>
> We added baseline comparisons (SLDS, rSLDS, mp-rSLDS) for predicting task-related variables. Results show that CBP consistently outperforms the baselines. However, **achieving competitive predictive performance is not the final objective. Our goal is to uncover the cognitive bases and connectivity patterns that drive these task variables, thereby providing transparent and scientifically grounded insights. (see revisions in Section 6 ⑤)**
>
> ### **(5) Improvements in clarity of presentation. (Reviewers VyUw, GVSN, pNLA)**
>
> (i) We added missing definitions of mathematical symbols and equations, and corrected terminology and grammatical issues.
>
> (ii) We polished the Introduction, Related Work, and Problem Description sections to better highlight the contributions and improve narrative clarity.
>
> (iii) We added qualitative explanations of the regularization terms. **(see revisions in Appendix R)**
>
> **References**
>
> [1] Camden J MacDowell and Timothy J Buschman. Low-dimensional spatiotemporal dynamics underlie cortex-wide neural activity. Current Biology, 30(14):2665–2680, 2020
>
> [2] Xiangmin Han, Rundong Xue, Jingxi Feng, Yifan Feng, Shaoyi Du, Jun Shi, and Yue Gao. Hypergraph foundation model for brain disease diagnosis. IEEE Transactions on Neural Networks and Learning Systems, 36(10):17702–17716, 2025.
>
> [3] Ziquan Wei, Tingting Dan, Tianlong Chen, and Guorong Wu. Brainmoe: Cognition joint embedding via mixture-of-expert towards robust brain foundation model. In The Thirty-ninth Annual Conference on Neural Information Processing Systems, 2025.
>
> [4] Noga Mudrik, Ryan Ly, Oliver Ruebel, and Adam S Charles. Creimbo: Cross-regional ensemble interactions in multi-view brain observations. The International Conference on Learning Representations, 2025.

---

### Author Response · Authors · 2025-12-02
**Summary of the Discussion Period**

# Summary of the Discussion Period

Dear Area Chairs and Reviewers,

We sincerely thank you for your additional time and efforts. We hope that the following summary provides a clear and fair overview of the current discussion.

Our work presents the interpretable, non–deep-learning framework for cross-cognitive-task brain modeling (CBP). **CBP uncovers a stable set of cognitive bases and connectivity patterns in the human brain, whose reliability is validated through systematic quantitative evaluations and a battery of perturbation analyses.** Leveraging these shared cognitive components, CBP generalizes effectively to both the HCP dataset and a multi-stage learning dataset, achieving accurate prediction of task states and learning outcomes while revealing the key cognitive connectivity patterns underlying these behaviors. Overall, CBP provides an interpretable framework for large-scale cognitive dynamics and offers mechanistic insight into cross-task cognition.

We are pleased that all reviewers acknowledged the importance of our scientific finding that human brain activity exhibits a set of task-invariant cognitive bases and connectivity patterns, as well as the high interpretability of the proposed model.

For reviewers **VyUw (Initial score: 2)** and **GVSN (Initial score: 4)**, we have addressed all their concerns with detailed point-by-point responses. ***Reviewers VyUw and GVSN ultimately decided to raise their scores.***

For reviewers **tXJt (Initial score: 6)** and **pNLA (Initial score: 6)**, we also provided complete responses and implemented all necessary revisions in the manuscript, further improving the overall quality of the work. However, due to the constraints of this special period, the reviewers were not able to further participate in the discussion.

In summary, we are grateful for all reviewers’ insightful comments, leading to a strengthened manuscript. The reviewers’ positive remarks and specific suggestions are documented in detail in our responses.

Sincerely,

Authors of ICLR Submission 6762

---

### Note · Authors · 2026-01-27

**Comment:**

We thank the Program Chairs, Area Chairs, and Reviewers for their evaluation.
In this work, we do not simply apply the modeling frameworks of dLDS/CREIMBO (Yezerets, Mudrik & Charles, 2025; Mudrik et al., 2024) to multi-task data. In the cross-session setting considered by CREIMBO, different sessions are treated as asynchronous and partial observations of the same underlying neural system, where variations across sessions primarily arise from sampling and observation conditions rather than systematic interventions on the system dynamics. Under this assumption, CREIMBO aims to capture shared dynamical structure across sessions by sharing the connectivity generation mechanism F.

In contrast, in multi-task fMRI settings, tasks correspond to distinct cognitive conditions that systematically modulate brain dynamics, resulting in substantial changes in activation patterns and interaction structures across tasks. Consequently, the core challenge of cross-task modeling is not to aggregate multiple observations, but to identify cognitive components and connectivity patterns that remain stable and reusable under task-dependent modulations. To address this challenge, our method explicitly introduces cross-task sharing and task-invariance constraints at the level of latent cognitive bases A, rather than sharing only the connectivity mechanism F. This modeling objective and constraint design differ fundamentally from the cross-session mechanism in CREIMBO, which does not explicitly enforce cross-task sharing or model task-invariant cognitive bases A.

**Withdrawal Confirmation:**

I have read and agree with the venue's withdrawal policy on behalf of myself and my co-authors.

---

### Meta-Review · Area_Chair_ex5e · 2026-01-07

**Summary:**

The goal of this work is to develop a time-varying linear dynamical system modeling framework for multi-task data. In particular, the model is heavily reflective of the dLDS framework cited (Mudrik et al. 2025) but trained over multiple tasks. This in and of itself is not novel since the modeling framework exists and the application to multi-task setting is somewhat dampened by the fact that dLDS does not assume at all that there is a single task. In fact in studies on C. elegans (Yezerets, Mudrik & Charles, 2025 and Mudrik et al. 2024), multiple "behaviors" of the system are easily captured. I think that's pretty straightforward since there seems to be no novelty on the model or training that differs from past work. This might be why the authors do not compare to dLDS in their results, but rather only switched systems and others. Reviewer VyUw did raise this point, and the authors responded by stating that there was novelty in the cross-task "mechanism", but to me this looks the same as the "cross session" mechanism in the Mudrik et al. 2025 paper. I'm not convinced in the current presentation that there is significant modeling novelty here.

The main contribution then is the application to the human connectome dataset, which is interesting and has some really nice elements, but is not exactly a fit to ICLR.

Other concerns raised were on the linearity assumptions, sensitivity to parameters, sensitivity to approximating nonlinear models with the (sparse) linear mixture model, and experimental validity/data processing. I find these to be secondary to the main novelty concern. Given the novelty issue and the borderline scores, I think that this work is not yet ready for publication.

**Reviewer Concerns:**

I think that the majority of concerns that were raised with respect to clarity on the experiments, and testing the model under more complex conditions were well addressed. As stated above, these were not as important as the main novelty concern which I feel that if there is such a difference, needs to be extremely clearly stated in the paper in terms of the novelty over prior work (including comparisons in the experiments).

**Reviewer Scores:**

The initial scores were 6,6,4,2. Of these the 2 and 4 indicated they would raise their score, with the Reviewer pNLA stating that they would like to raise their 4 to a 6, and Reviewer VyUw stating that they would raise their score but not specific as to what.

---

### Decision · Program_Chairs · 2026-01-26

Reject